# Condensation Reactions of 2-Aminothiophenoles to Afford 2-Substituted Benzothiazoles of Biological Interest: A Review (2020–2024)

**DOI:** 10.3390/ijms26125901

**Published:** 2025-06-19

**Authors:** Itzia I. Padilla-Martínez, Alejandro Cruz, Efrén V. García-Báez, Jessica E. Mendieta-Wejebe, Martha C. Rosales-Hernández

**Affiliations:** 1Laboratorio de Química Supramolecular y Nanociencias, Departamento de Ciencias Básicas, Unidad Profesional Interdisciplinaria de Biotecnología, Instituto Politécnico Nacional, Av. Acueducto s/n, Colonia Barrio La Laguna Ticomán, Ciudad de México 07340, Mexico; ipadillamar@ipn.mx (I.I.P.-M.); egarciaba@ipn.mx (E.V.G.-B.); 2Laboratorio de Biofísica y Biocatálisis, Sección de Estudios de Posgrado e Investigación, Escuela Superior de Medicina, Instituto Politécnico Nacional, Av. Salvador Diaz Mirón esq. Plan de San Luis s/n, Casco de Santo Tomas, Miguel Hidalgo, Ciudad de México 11340, Mexico; jmendieta@ipn.mx (J.E.M.-W.); mrosalesh@ipn.mx (M.C.R.-H.)

**Keywords:** condensation, 2-aminothiophenol, carbonyl compounds, benzothiazoles, biological activities

## Abstract

Several benzothiazole (BT) derivatives have recently been explored in medicinal chemistry, and they are frequently reported in the literature. The interest in this kind of heterocyclic compounds and their structural hybrids has been increasing, as shown by several reviews reported over the last decade. In this context, we found that about 70 articles related to the synthesis of BT derivatives that studied their biological activities were published in the last five years. From this, we prepared a review on the synthesis and biological activity studies about this topic. In this bibliographic review it was found that medicinal chemists also explore BT derivatives in search of anticancer and anti-Alzheimer’s candidates. This review comprehends 70 articles, published between 2020 and 2024, related to the synthesis of BT derivatives with the purpose of assessing their biological activities. On the other hand, BT derivatives have been explored as molecular species that perform two or more biological actions, called multifunctional drugs. Some accounts related to the structure–activity relationship which provide a framework for drug discovery and design are also discussed. The synthetic methods of BT synthesis include the use of biocatalysts, solvent-free conditions, photocatalysts, and catalysts supported on nanoparticles. Studies also explore renewable energy sources such as microwave, UV, and visible-light and mechanochemical sources.

## 1. Introduction

Benzothiazole (BT) is an aromatic heterobicyclic system containing a benzene ring fused to a thiazole ring. This small molecule is frequently present in compounds that possess a wide range of biological activities. BT derivatives are rarely found in natural products; however, some of them have been found in terrestrial and marine natural products as part of simple or more complex structures. For instance, in the 1940s, the simple firefly *d*-luciferin structure [1,2,3], the more complex rifamycins P and Q, and thiazinotrienomycin F and G were isolated (Figure 1) [4,5]. Aromatic constituents of cranberries and tea leaves and flavor compounds such as the benzothiazole (BT) alkaloids nordercitine, dercitamide, dercitamine, and cyclodercitin, from the *Dercitus* sp. and *Stellata* sp. sponges, were also isolated (Figure 1) [6,7]. On the other hand, since riluzole (6-trifluoromethoxy-2-benzothiazolamine) has been clinically used to diminish amyotrophic lateral sclerosis progression and as an anticonvulsant drug [8,9,10], medicinal chemists have been interested in developing synthetic methodologies to afford BT derivative compounds.

From these findings, BT cores have been found in several synthesized drugs. From 1970 to the present, BT derivatives have been found to possess antiviral [11,12,13], antibacterial [13,14,15,16], antimicrobial [17,18,19], fungicidal [20,21], antiallergic [22,23,24], antidiabetic [25,26,27], antitumoral [28,29,30,31], anti-inflammatory [32,33], and anthelmintic [34,35,36] properties and have recently been shown to possess anticonvulsant and antioxidant [37,38,39,40,41,42] biological activities. For instance, zopolrestat has been used since 1991 in treating diabetes (Figure 2a) [43], compound 2-(3,4-dimethoxyphenyl)-5-fluoroBT (PMX 610, NSC 721648) has shown antitumor activity (Figure 2b) [44,45], and the Schiff bases shown in Figure 2c were found to be useful in the treatment of Alzheimer’s disease [46] (Figure 2).

On the other hand, the BT nucleus has been found in molecules used as ligands for catalysis [47,48]. In this context, we found that the investigation of the synthesis and biological activity of BT derivatives in medicinal chemistry has increased. In the last five years we found twenty-one article reviews about this topic, from 2020 to 2024 [49,50,51,52,53,54,55,56,57,58,59,60,61,62,63,64,65,66,67,68,69,70], several of them related to the use of BT derivatives as anticancer agents [51,52,53,54,55,56,57,58,62,63,64]. Also, we found in this period about 53 articles related to the condensation reaction of 2-aminothiophenol (ATP) to synthesize BT derivatives. In some of these, the synthesized compounds were studied concerning their biological activity. By conducting this search, we prepared a bibliographic review on the synthesis and biological activity studies about this topic. In this review it was found that medicinal chemistry also explores BT derivatives in search of anticancer [53,54,55,56,57], anti-Parkinson’s [71,72], and anti-Alzheimer’s candidates [73,74,75]. In addition, some accounts of the structure–activity relationship that provide a framework for drug discovery and design were also discussed. On the other hand, BT derivatives were also explored as molecules capable of two or more biological actions, known as multifunctional drugs [76]. Some recent examples of this kind of BT-based compounds that have been clinically approved are halethazole (antibacterial) [77], thioflavin-T (amyloid imaging agent) [78], dimazole (antifungal) [79], ethoxzolamide (treatment of glaucoma) [80], phortress (antitumoral) [81], riluzole (antidepressant) [82], flutemetamol (radiopharmaceutical) [83], and frentizole (immunosuppressive) [84] (Figure 3).

## 2. Condensation of 2ATPs with Carbonyl Compounds

One of the best-known reactions that affords 2-substituted benzothiazoles (BTs) is the condensation of carbonylic compounds with 2-aminothiophenoles (2ATPs). The followed pathway consists of three stages: (a) imine thiophenol (ITP) formation, (b) cyclization to benzothiazolidine (BTI), and (c) reduction to BTs (Figure 1). These reactions have been carried out with and without the use of catalysts.

### 2.1. Condensation with Aldehydes

In one step, the condensation reaction of 2ATP with several benzaldehydes using sodium hydrosulfite (Na_2_S_2_O_4_) as an oxidizing agent in refluxing a mixture of water–ethanol as the dissolvent for 12 h was carried out to afford 2-arylBTs **1a**–**k** in 51–82% yields (Table 1) [85]. On the other hand, the condensation of 5-substituted 2ATPs as hydrochlorides or disulfides with benzaldehydes under refluxion for 36 h afforded the BTs **2b**–**k** in 53–82% yields and **3a**–**k** in 51–84% yields.

Compounds were evaluated as sun-filtering, antioxidant, antifungal, and antiproliferative agents [85]. Compound **1g** was the most effective blocker of hERG potassium channels expressed in HEK 293 cells, resulting in 60.32% inhibition, with an IC_50_ = 4.79 μM. Compounds **1g**, **1h**, **1k**, **2d**, **2g**, **3d,** and **3g** were considered broad-spectrum and ultraviolet A filters (UVA filters) because their λc ≥ 370 nm and the ultraviolet A-protector factor (UVA-PF) values were greater than one-third of their sun protection factor (SPF) values. Compounds **1d** and **1j** were regarded as candidates for UVB protection. All compounds were photostable, especially **2e,** which showed a degradation rate of less than 2%.

It was found that the wavelength of maximum absorption (λ_max_) was related to stereo electronic effects by substituents on the 2-aryl or BT moiety, which alter the electron density of compounds. The -OH group at the para position of the 2-aryl group leads the resonance interaction of the lone pair of oxygen and the π cloud of 2-aryl and the BT group. This shift increases if -OH is at the ortho position, due to the formation of an intramolecular hydrogen bond between the nitrogen of BT and the hydrogen of the -OH group. On the other hand, the order of λ_max_ along the series was -H ≪ -SO_2_NH_2_ < -COOH. Additionally, the -COOH or -SO_2_NH_2_ groups on the BT ring increased the photostability of the compounds.

Compounds **1i**,**j**, **2g**,**h**, and **3r**,**s** showed a good antioxidant profile. Compound **1h** had the highest antifungal activity against the dermatophytes *Trichophyton tonsurans*, *Epidermophyton floccosum*, and* Microsporum gypsum*. Significant activities were found for compounds **1f** and **1g** against *Epidermophyton floccosum *and* Microsporum gypseum. *Compounds **1g** and **1k** exhibited high antiproliferative activity against CEM and SK-Mel 5 tumor cells, **1g** (6-fold) and **1k** (25-fold) being more selective for CEM and SK-Mel 5 cells, respectively. Because of their various biological activities, some 2-arylBTs could be applied as drugs in the treatment of neoplastic diseases such as melanoma, childhood leukemia, and pancreatic cancer. However, based on the experimental results and SAR studies on positions 2 and 6 of 2-arylBTs, compounds **1g** and **1k** were proposed as powerful candidates for designing multifunctional drugs.

2ATP was condensed with substituted benzaldehydes by using biodegradable rice husk chemically activated carbon (RHCAC) as the catalyst at room temperature in ethanol–water to afford 2-arylBTs **4a**–**f** in 93–98% yields (Table 2) [86]. The catalyst was recovered by filtration, washed, dried, and recycled for eight runs, unlike a corrosive and toxic acid catalyst that was difficult to recover and separate.

No significant effects of electron donation were observed, while OMe, CH_3_, and Cl groups and electron-withdrawing NO_2_ groups were found, and the reactions were completed in short times (5–10 min), giving high yields of the desired products (93–98%).

A series of 2-arylBTs **5a**–**f** were synthesized from the condensation reaction of 2ATP with substituted benzaldehydes in 1,4-dioxane under an O_2_ atmosphere in the presence of [PhI(OH)OTs] (Koser’s reagent) as a catalyst at room temperature (Table 3) [87]. The features of this protocol included short reaction times (15 min), a broad substrate scope, 80–90% yields, low catalyst loading, and scalability.

Six *p*-substituted 2-arylBTs **6a**–**f** were obtained in 70–90% yields from the condensation of 2ATP with *p*-benzaldehydes catalyzed with Cu_2_O and DMSO as oxidants in string at room temperature for 3–5 h (Table 4) [88]. The reaction tolerated a wide range of functional groups. The compounds **6c** and **6e** showed equal antifungal activity against *A. niger* and the highest activity against *C. albicans*, compared to voriconazole as a reference.

Three green synthetic protocols were used to synthesize 6-substituted 2-arylBTs, **7a**–**f** and **8a**–**f**, in 62–89% yields (Table 5) [89]. Method A: A stirred suspension of 4-substituted-2ATP and the 2-hydroxybenzaldehyde was heated in glycerol at 110–170 °C for 1 to 24 h to afford compounds **7a**–**c** in 62–71% yields. Method B: A stirred suspension of the 5-substituted 2ATP disulfide and 2-hydroxybenzaldehyde or 2-methoxybenzaldehyde was heated in glycerol at 160–175 °C for 30–60 min to afford compounds **7a**–**c** in 74–80% yields and **8a**–**c** in 70–79% yields. Method C: The corresponding 2-substituted benzaldehyde was added to a stirred solution of 4-amidino-2ATPate in glacial acetic acid, and the mixture was heated under nitrogen for 4 h, then alkalinized with NaOH (pH 10–11); the resulting free base was mixed with 2-propanol and methanesulfonic acid and stirred at room temperature for 2 h to afford compounds **7d**, **7e**, **8d**, and **8e** in 62–83% yields. Method D: A solution of SnCl_2_ dihydrate in conc. HCl and methanol was added to the corresponding 6-nitro-substituted BTs **7b** and **8b**, then refluxed for 30 min. The mixture was alkalinized with 20% NaOH (pH 8–9) to afford compounds **7f** and **8f** in 89.3 and 81.6% yields, respectively. The influence of the hydroxy and methoxy groups and substituents on C6 on the 2-arylBT scaffold against antibacterial, antitumor, and antioxidant activities was studied. The amidino derivatives **7d** and **8d** showed modest activity against Gram-negative and Gram-positive bacterial strains. At the same time, compounds **7b**, **7c**, **7e**, **7f**, and **8f** resulted in potent and selective antiproliferative activity towards tumor cells, without activity against human skin fibroblasts. Compounds **7f** and **8f** were the most selective against the growth of HeLa cell lines. Compound **7f** resulted in the most promising radical scavenging activity. HIF-1α hydroxylated protein was upregulated by treatment of HeLa cells with compounds **7f**, **8c**, and **8f**, which were considered potent suppressors of the hypoxia-induced HIF-1 protein. Compounds **7f** and **8f** were proposed to lead compounds for further rationalized design of the BT skeleton. Compound **7f** with the OH group on the 2-arylBT core had the most promising antioxidative activity as evaluated by DPPH, ABTS, and FRAP in vitro assays. The presence of the amino protonated group attached at the BT moiety was essential for the antiproliferative and antioxidant activities observed, exerted through a change in the levels of the reactive oxygen species-modulated HIF-1 protein. The BTs **7a**–**f** and **8f** showed good antioxidant activity (38 to 117 µM), while BTs **8a**–**e** had a low ability for stabilization of ABTS^•+^ radicals (>200 µM).

In a two-step protocol, the corresponding *p*-bromophenyl-ITP intermediate, accessed from the condensation of 2ATP and *p*-bromobenzaldehyde, suffered a Pd-catalyzed cyclization to 2-bromophenyl-BT **9** in [BMIM][BF_4_] or [BMIM][PF_6_] as a recycled IL solvent. Compound **9** was functionalized by cross-coupling reactions (Suzuki, Heck, or Sonogashira) and catalyzed by Ni or Pd to afford a series of substituted 2-aryl-/heteroaryl-BTs, **10a**–**e**, **11a**–**d**, and **12a**–**d**, in acceptable yields (55–86%) under mild reaction conditions (Figure 2) [90]. The yields were from moderate to good for **10a**–**e** (60–81%, 12–14.5 h), **11a**–**d** (63–80%, 16.5–17 h), and **12a**–**d** (55–80%, 14.5–16 h).

2-substituted BTs, **14a**,**b** (25, 34%) and **17a**,**b** (52, 60%), were designed and synthesized from BTs **13** (60%) and **15** (34%), which were obtained from the condensation of 2-aminobenzothiazole (2ABT) with 3-bromobenzaldehyde and lactic acid, then oxidation of the derived alcohol with MnO_2_, to be tested as antiproliferative cancer cell lines (Figure 3) [91]. Compounds **217a**–**b** were obtained from compound **15** by (a) cyclization of 2ABT with 3-bromobenzaldehyde to afford compound **13**, and the BTs **14a**,**b** were obtained from compound **13** by (b) Buchwald–Hartwig amination with anilines. Compounds **17a**–**b** were obtained from (d) α-bromination with CuBr_2_, then (e) reaction with the corresponding substituted thiourea. Antiproliferative assays using cancer cells from the breast (MCF7) and prostate (22Rv1 and PC3) were carried out for all synthesized BTs. However, these compounds had worse activity than JG-98 as a reference (IC_50_ values from ~0.7 to 13 µM).

A photo-assisted radical cyclo-condensation of 2ATP with a series of aromatic and aliphatic aldehydes was designed using wosin Y as a photocatalyst and K_2_CO_3_ or Et_3_N as basic media and *tert*-butyl hydroperoxide (TBHP) (70% in water) in acetonitrile to afford the 2-aryl(alkyl)BTs **18a**–**j** in 70–92% yields (Table 6) [92]. Unfortunately, the designed method required harsh reaction conditions, a long reaction time (24–36 h), and flash column chromatography and allowed no recovery of the catalyst.

A reasonable mechanistic process has been proposed beginning with the in situ formation of the corresponding imine and the excitation of eosin Y under blue LED irradiation, then the reaction of eosin Y* with TBHP to generate an (eosin y)**^.^**-radical anion and a *t*-Bu^.^ radical, which undergo hydrogen atom abstraction with the imine to produce the Ar-O**^.^** imine-fenoxy radical. Then, there occurs an intramolecular cycloaddition to the corresponding benzothiazoline, which is transformed to a benzothiazoline**^.^**+ radical cation that reacts with the eosin Y**^.^**-radical anion to afford the corresponding BT. This method tolerates several aldehydes, including aliphatic, aromatic, cyclic, and heteroaromatic aldehydes, but has several limitations, such as the requirement of a transition metal, a high temperature, and a pre-synthesized catalyst/ligand scaffold.

As previously reported, three series of substituted 2-arylBTs, **21**–**23**, were synthesized as VEGFR-2/FGFR-1/PDGFR-β multi-angiokinase inhibitors targeting breast cancer (Figure 4) [93]. The compounds **21** and **22** were obtained from the condensation of 2ATP and vanillin in refluxing DMF to afford the 2-arylBT **19**, which was subsequently reacted with methyl bromoacetate, followed by hydrazinolysis with hydrazine hydrate in refluxing ethanol to afford the hydrazide **20**, which through acid catalysis with aldehydes or ketones afforded the Schiff bases **21a**–**n** and **22a**–**c** in 40% to 88% yields. On the other hand, the reaction of hydrazide **20** with isocyanates and isothiocyanate in refluxing ethanol afforded the corresponding ureas **23a**,**b** in 84 and 72% yields and thiourea **23c** in a 61% yield.

Compounds **21d**, **21f**, **21i**, and **21k** showed high inhibitory activity against multiple kinases: VEGFR-2 (IC_50_s of 0.19, 0.18, 0.17, and 0.13 μM, respectively), FGFR-1 (IC_50_s of 0.28, 0.37, 0.19, and 0.27 μM, respectively), and PDGFR-β (IC_50_s of 0.07, 0.04, 0.08, and 0.14 μM, respectively). Additionally, the substituted 2-arylBTs exhibited promising cytotoxic activity against the MCF-7 cell line comparable to that of sorafenib as a reference drug (IC_50_ = 4.33 μM). The most potent arylBTs were **21d** and **21i** (IC_50_s of 7.83 and 6.58 μM). Also, **21d** and **21i** had VEGFR-2 inhibitory activity in MCF-7 cells of 81% and 83% compared with sorafenib (88%). Molecular docking of compounds in the VEGFR-2 and FGFR-1 active sites showed that the 2-phenylBT moiety was placed in the hinge region of the tyrosine kinase (RTK)-binding receptor site. In contrast, the amide moiety showed hydrogen bonding interactions with the amino acids, directing the aryl group to the hydrophobic allosteric back pocket of the RTKs in a type II-like binding mode. Additionally, the aryl BTs exhibited good ADME properties, facilitating further optimization in drug discovery.

An acetic acid-mediated three-component condensation reaction of 2ATPs and *α*, *β*-unsaturated aldehydes in the presence of thiophenols was developed to afford 2-thioalkyl BTs **24a**–**t** (Figure 4) [94]. Metal-free conditions were used in the reaction, with oxygen as an oxidant. This method was regioselective, tolerant of several functional groups, and provided access to various kinds of BTs with modest to good yields (34–81%).

On comparing the yields of compounds **24a** (70%), **24b** (42%), and **24d**, the position of the methyl/phenyl group (R^4^) on the *α*, *β*-unsaturated aldehydes affected the reaction yield. However, *α*, *β*-unsaturated aldehydes bearing an electron-donating group or electron-withdrawing group like Me, Cl, or Br at the para position of the phenyl ring produced the compounds **24e**–**g** in moderate yields (54–57%). Also, moderate yields (51–59%) were observed when Me, Cl, and F substituents were located at different positions on the benzene of the BT ring. Thiophenols containing methyl groups at the para, meta, or ortho positions did not show an influence and efficiently reacted in moderate yields. However, electron-donating groups such as *^t^*Bu and OMe at the para position of thiophenols afforded BTs in 71% (**24k**) and 80% (**24l**) yields, respectively. Halogen-containing thiophenols such as F, Cl, and Br successfully afforded the thioalkyl-substituted BTs **24a**, **24m**, and **24n** in 53–70% yields. Various disubstituted thiophenols resulted in good substrates for this reaction to afford the BTs **24o**–**q** in 56, 80, and 81% yields, respectively.

The eco-friendly biocatalytic oxidant system Laccase/DDQ (2,3-dichloro-5,6-dicyano-1,4-benzoquinone) was applied to condense 2ATP with aromatic aldehydes to afford 2-arylBTs **25a**–**x** in 65–98% yields using O_2_ pure or from the air as an oxidant in aqueous media at room temperature in 1 h (Table 7) [95]. Two steps were required in the aerobic oxidative cyclization: (1) chemical cyclization and (2) chemoenzymatic oxidation. Benzaldehydes with electron-donating and electron-withdrawing groups, heterocyclic and α,β-unsaturated aldehydes, 1-naphthaldehyde, 2-naphthaldehyde, 9-anthraldehyde, and terephthaldehyde were successfully applied to prepare the corresponding BTs. The advantages of this method are as follows: (1) the use of air or O_2_ as an environmentally benign, inexpensive, and abundant oxidant agent and the formation of H_2_O as a non-toxic by-product; (2) the synthesis of BTs in good to high yields in aqueous media at room temperature; (3) its superiority with respect to other available methods and its attractiveness to the pharmaceutical industry owing to its being free from any toxic and expensive transition metals and halide catalysts; (4) its conforming to several of the guiding principles of green chemistry.

In a one-pot reaction, 2ATP was condensed with benzaldehydes and aromatic heterocyclic aldehydes under solvent-free conditions using ZnCl_2_ nano-flakes supported on nano-hydroxyapatite (HAp) as a catalyst to afford the 2-arylBTs **26a**–**l** in 65–95% yields in 15–90 min (Table 8) [96]. The reaction was fast, eco-friendly, and efficient. The molecular docking validation with transpeptidase and 14 α-demethylase enzyme inhibitors showed that five 2-substituted-BTs gave good energy values. Also, antioxidant studies showed that four BTs were promising in relation to ABTS (inhibition %) compared with ascorbic acid. Substituted and disubstituted benzaldehydes with electron-donating groups on R^2^, such as 4MeO and OH groups, and electron-withdrawing groups, such as F, Cl, and NO_2_ groups, gave BTs **26a**,**b**,**d**–**h** with excellent yields (75–80%); however, 1,2,3-OMe-substituted and 3-pyridil-substituted benzaldehydes gave BTs **26c** and **26j** in 67 and 65% yields, respectively.

The condensation reaction of substituted 2ATPs with several aromatic and aliphatic aldehydes was catalyzed with sulfated tungstate (ST) under solvent-free, room temperature, and ultrasound irradiation conditions to afford 2-substituted BTs, **27a**–**z** and **28a**–**d**, in excellent yields (90–98%) (Table 9) [97]. ST is considered a mildly acidic, easy-to-prepare, non-toxic, recyclable, efficient, and heterogeneous green catalyst. This method is very good for the synthesis of 2-substituted-BTs from several functionalized aromatic, aliphatic, and heteroaromatic aldehydes. The advantages of this method are as follows: easy handling, functional group compatibility, short reaction times (5 min), high catalytic activity and recyclability, chemoselectivity, very good yields, no column purification, low corrosiveness, and environmental compatibility.

2ATP and substituted aryl aldehydes were condensed using ruthenium silicate (RS-1) zeolite as a catalyst in a hydrothermal process to afford the 2-arylBTs **29a**–**i** in 85–93% yields (Table 10) [98]. The benefits of this protocol include mild reaction conditions, short reaction times (30 min), and high thermal catalyst and catalysis recyclability.

Substituted BTs **30a**–**t** were synthesized in 82–94% yields in 3 h by the visible-light-induced condensation–cyclization reaction of substituted 2ATP with substituted aromatic aldehydes (Table 11) [99]. Fluorescein was used as a photocatalyst, a blue LED lamp was used as the light source, and atmospheric molecular oxygen was required for the reaction to proceed.

Substituted aromatic aldehydes, such as 4-Br, 4-Cl, and 4-F (electron-withdrawing) and 2-Me, 4-Me and 4-OMe (electron-donating) substituents, afforded BTs in high to excellent yields: **30d**–**f** (88–94%) and **30g**–**i** (87–92%). Notably, the position of the substituents on the aromatic aldehydes had little effect on the yield of the BTs **30b**–**d** (88–91%) and **30g**,**h** (91, 92%). In addition, two electron-donating groups on the aromatic aldehydes gave a lower yield (82%) of the BT **3j**. The reactions were smooth, with good functional group tolerance. Additionally, the catalytic system eliminates the need for an oxidant or metal catalyst, aligning with the principles of green chemistry.

The 2-aryl-BTs **31a**–**t** were synthesized in 64–99% yields in 10 min by combining enzymatic (trypsin) and visible-light (450 nm) catalysis under an air atmosphere (Table 12) [100]. The method consisted of biocatalytic condensation of 2ATP with aromatic or aliphatic aldehydes to afford the corresponding benzothiazoline **BTI** as an intermediate. Subsequent visible-light-induced oxidization produced the 2-arylBTs in approximately 10 min. Additional oxidants or metals were not required in this protocol, which was environmentally benign.

It was found that the electronic effect on the substituents on the benzene ring influences the reaction. Aldehydes containing electron-deficient groups afforded the BTs **31a**, **31g**, and **31j**, which were obtained in 98–99% yields, while an electron-rich pair group afforded the BT **31m** in 75% yields. If the electron-richness group of the aldehyde is increased, the yield of BTs decreases, as in the series of BTs **31g**–**i**, whose yields were 98%, 95%, and 89%, and the yield in BT **31p** was reduced to 64%. In the case of steric hindrance of methyl-substituted benzaldehydes, the yield is negligible, as in the BTs **31c**–**e** (96–98%). With a larger volume of 1-naphthaldehyde, the yield of the BT **31s** decreases to 78%.

An ionic liquid immobilized on silica-coated cobalt–ferrite magnetic nanoparticles (CoFe_2_O_4_@SiO_2_@PAF-IL) was formed. This hybrid nanostructure was used as a catalyst in the condensation reaction of 2ATP with aromatic aldehydes to synthesize the 2-aryl-BTs **32a**–**i** in 83–91% yields (Table 13) [101]. The reaction was carried out under heating in solvent-free conditions at 70 °C for 10 min. The advantages of this procedure included the use of solvent-free conditions, the simple work-up, the short reaction times, and the environmentally benign conditions. The nano-catalyst could be reused several times without loss of catalytic activity and was easily separated.

Substituted 2ATP disulfides or dihydrochlorides were condensed with the corresponding aldehydes in refluxing glycerol without a catalyst for 45 min to afford the previously designed 6-cyano- (**33a**, 83%; **33d**, 64%) and 6-amidino- (**33b**,**c**,**e**,**f**, 22–56%) 2,5-disubstituted furane-BTs **33a**–**f** to be screened for antimicrobial and antitumor activities (Figure 5) [102]. The antitumor activity was tested on the human lung cancer cell lines A549, HCC827, and NCI-H358, with MTS cytotoxicity and in vitro BrdU proliferation assays performed using 2D and 3D cell culture methods. Compounds **33b**,**c**,**e** resulted in possible antitumor activity to stop the proliferation of cells. Broth microdilution testing was used to evaluate the antimicrobial activity on Gram-negative *E. coli* and Gram-positive *S. aureus* and *S. cerevisiae* as eukaryotic model organisms, according to Clinical Laboratory Standards Institute (CLSI) guidelines. The BTs **33b**,**c**,**e** and 233f showed promising antibacterial activity. All compounds were more activated in the 2D than in the 3D assays on the three cancer cell lines and in the antimicrobial assays. Compounds with the amidine group at the 6-position of the benzothiazole ring, **33b** and **33e**, gave low yields (22 and 28%, respectively).

Compound 4-(BT-2-yl)-2-methoxyphenol **34**, obtained from the condensation of 2ATP with the corresponding benzaldehyde, was transformed to compound 2-[4-BT-2-yl)-2-methoxyphenoxy]acetohydrazide **36** through ethyl 2-(4-BT-2-yl)-2-methoxyphenoxy) acetate **35**, then reacted with chlorine acetyl chloride to afford the 2-(4-BT-2-yl)-2-methoxy-phenoxy)-N′-(2-chloroacetyl)acetohydrazide **38** (Figure 5) [103]. The acetohydrazides **36** and **38** were reacted with anhydrides and amino acids to afford the BT derivatives **37a–f** in 65–90% yields and **39a–k** in 65–90% yields, respectively. The in vitro anticancer activity of all the compounds exhibited promising potency against hepatocellular carcinoma HepG-2 (IC_50_s from 0.7 ± 0.4 to 1.0 ± 0.7 μM) and very good potency against breast cancer cells MCF-7 (IC_50_s from 2.5 ± 2.5 to 3.5 ± 3.4 μM) compared with the standard drug doxorubicin (IC_50_s = 1.0 ± 0.8 μM and 2.9 ± 1.9 μM, respectively). The highest cytotoxic activity was observed for compounds **37a**–**c**, **39c**, and **39f**. Also, these compounds were further evaluated for their EGFR inhibitory activity compared with the reference drug erlotinib. Molecular docking studies of the promising compounds **39a**–**c** were carried out to interpret their enzymatic activities. Moreover, compounds **39a** and **39b** exhibited considerable pre-G1 and G2/M cell cycle arrest compared with the untreated MCF-7 cells. Additionally, **39a** and **39b** increased the levels of Bax, p53, and caspase-3 levels while decreasing the levels of Bcl-2, which are oncogenic parameters. The results of these BT-based derivatives proposed an excellent framework for detecting new potent antitumor leads.

The condensation reaction of (un)substituted 2ATP, 2ATP-disulfide, or 2ATP-hydrochloride with furfural, pyrrole-2-carboxaldehyde, or 2-thiophenecarboxaldehyde occurred under stirring in ethanol at room temperature, then unsubstituted 2ATP was reacted under refluxion for 12 h and substituted under refluxion for 36 h in the presence of an aqueous solution of sodium hydrosulfite as a catalyst to afford the 2-substituted-BTs **40a**–**i** in 69, 48, 87, 34, 30, 48, 31, 29, and 54% yields (Figure 6) [76]. The synthesized BTs were investigated for their photoprotective, antioxidant, antiproliferative, and antifungal activities. Compounds **40d** and **40g** exhibited a multifunctional profile with an excellent filtering UVB capacity, which was higher than that of PBSA, which served as a reference and is currently used as a UV sunscreen filter. These compounds were the best at inhibiting the growth of dermatophytes and *C. albicans*, and **40g** showed good antioxidant activity. Furthermore, **40d** was effective on melanoma tumor cells (SK-Mel 5). These compounds are proposed as new skin preventive and protective agents.

2ATP was cyclo-condensed with a series of aromatic/aliphatic aldehydes in the presence of bovine serum albumin (BSA) as a biocatalyst in water under stirring at room temperature for 8 h to afford a series of 2-substituted BTs, **41a**–**v**, in 79–93% yields (Table 14) [104]. The advantages of this protocol were its excellent yields, high atom economy, gram scalability, operational simplicity, and recyclability.

The condensation–cyclization reaction of 2ATP with aromatic benzaldehydes was carried out in a self-neutralizing acidic CO_2_–alcohol system to afford the BTs **42a**–**n** in 55–87% yields. Practicality, economic viability, and environmental friendliness were the advantages of this protocol (Table 15) [105]. The alkyl carbonic acid formed CO_2_, and methanol was proposed as an intermediate in the reaction mechanism, generating hydrogen cations to catalyze the reaction. 2-PhenylBT **42a** gave the highest yield (87%), while in the case of the 2-substituted BTs **42b**–**m** the yields decreased to 55–62%. However, in 5-chloro-2-phenylbenzothiazole **42n** the yield increased to 72%.

2ATP was condensed with aromatic aldehydes in the presence of 0.05 mol% of bacterium-derived hemoglobin *Vitreoscilla* (VHb) as a biocatalyst, *tert*-buthyl hydroperoxide (TBHP) as an oxidant, and dimethyl sulfoxide (DMSO) as a solvent at room temperature to afford 2-arylBTs **43a**–**n** in only 5 min with excellent yields (85–97%) (Table 16) [106]. The advantages of this protocol were its mild reaction conditions and high efficiency, with a wide substrate scope.

It was found that electron-rich groups on R^2^ resulted in decreased yields (**43b**,**c**,**g**,**i**; 92, 88, 89, and 87% yields, respectively) compared with 2PhBT (**43a**), whereas electron-poor moieties within the aromatic aldehydes (**43d**–**f**,**h**,**j**; 91–97%) were the best substrates. Aldehydes with heteroaromatic groups produced the BTs **43k**–**m** in high yields (93–95%) with a longer reaction time (20 min). The yield decreased significantly when the benzaldehyde contained the naphthalyl group, affording the BT **43n** in an 85% yield, due to the steric effect.

Substituted 2ATPs were condensed with a series of aldehydes at 80 °C using an environmentally friendly, inexpensive, and easily accessible Zn(OAc)_2_·2H_2_O (5 mol%) in a solvent-free reaction to afford a series of substituted 2-arylBTs, **44a**–**p**, in 67–96% yields in 30–60 min (Table 17) [107]. The reaction involving heterocyclic aldehydes interestingly afforded the corresponding BTs **44m**–**o** in good yields (79–84%), and with alkyl aldehydes, such as isobutyraldehyde, the corresponding BT **44p** was produced in a moderate yield (67%). The BTs **44a**–**o** were obtained in 79–96% yields.

The nanocomposite MNPs-phenanthroline-Pd (Fe_3_O_4_@SiO_2_-diPy-Pd) was synthesized to be used as a catalyst for the cyanation of aryl halides at 80 °C for 6 h in the synthesis of BTs **45a**–**h** in 89–98% yields (Table 18) [108]. The optimal conditions for the reaction were 20 mg of the MNPs-phenanthroline-Pd nano-catalyst and K_2_CO_3_ in DMF under reflux conditions. The catalyst was highly efficient and easily recovered and reused, without losing the morphology and dispersion of the particles. Aryl halide compounds with electron-donating groups at the *p*-position of the aryl group afforded the BTs **45c**,**d**,**f** in higher yields (96%, 98%, and 94%) than that of aryl with electron-withdrawing groups (BT **45e**, 92% yield), whereas aryl bromide in the *m*-position decreased the yield (BT **45h**, 89% yield).

The eco-friendly DABCO-based dicationic acidic ionic liquid [C_4_H_10_-DABCO][HSO_4_]_2_ (DABCO = 1,4-diazabicyclooctane) was synthesized, characterized, and successfully used as a catalyst in the condensation of aryl aldehydes under H_2_O reflux for 30–50 min to afford 2-phenylBTs **46a**–**h** in 85–91% yields (Table 19) [109]. The advantages of this protocol were its short reaction time; easy work-up; moderate to excellent yields; the use of green solvents; the use of non-metal, inexpensive catalysts; and catalyst recyclability. Aldehydes with electron-withdrawing groups at the *m*- or *p*-position of the aryl group afforded BTs **46b**,**e**,**f** in higher yields (87%, 90%, and 91%) than that of aldehyde with an electron-donating group (BT **45c**, 82%).

Six 1-aryl-4-BTyl-1,2,3-triazoles **47a**–**f** were prepared from condensation of 2ATP with aldehydes (Figure 7) [110]. The synthesized donor–acceptor molecules exhibited optical properties based on charge transfer emission depending on the substituent in the 1,2,3-triazole moiety. Compounds **47a** and **47e** with moderately electron-donating groups (CH_3_ and OCH_3_) or 4**7b** and **47c** with electron-withdrawing substituent groups (F and CF_3_) in the triazole fragment were blue-shifted, in contrast to compounds **47d** and **47f** with strong electron-withdrawing (-NO_2_) and electron-donating (-N(CH_3_)_2_) substituents, respectively, which showed red-shifted maxima. Compounds **5a**–**d** had low fluorescence quantum yields, **5d** being almost non-emissive with a ΦF less than 0.01 due to the –NO^2^ group. However, compound **5f** had the highest quantum yield, which increased with the decreasing polarity of the solvent.

Some of these compounds were active in human A2780, HeLa, and A549 cancer cell lines, exhibiting IC_50_ values in the low micromolar range and low cytotoxicity in healthy CHO cells. Interestingly, compound **47f** showed cytoplasmic staining determined by confocal fluorescent microscopy.

The fluorescent probe **51** based on BT was synthesized in a 53% yield by condensing 2ATP with 5-methyl salicylaldehyde in DMF and Na_2_S_2_O_5_ as an oxidant to afford BT **48** in an 80.5% yield, followed by three steps through BT **49** in a 62.5% yield and **50** in a 72% yield (Figure 6) [111]. The excellent selectivity enabled the detection of Cu^2+^. The probe exhibited a cyan blue fluorescence emission peak at 487 nm under 360 nm UV excitation with fluorescence quenching after adding Cu^2+^. The detection limit of Cu^2+^ of the probe was determined as 3.08 × 10^−7^ M. According to a Job plot and high-resolution mass spectrometry, the complexation ratio of the probe to Cu^2+^ was 2:1. An excited intramolecular proton transfer (ESIPT) was confirmed as the sensor mechanism. Under alkaline conditions, the probe showed cyan blue and green fluorescence under acidic conditions. On the other hand, due to the excellent membrane permeability and low cytotoxicity, the probe could detect Cu^2+^ in water samples and recognize Cu^2+^ in human breast cancer cells (MCF-7).

2ATPolate, 2ATP-disulfide, or 2ATP-hydrochloride were condensed with the corresponding aldehyde using two methods to afford the series of benzo[b]thienyl and 2,2′-bithienyl-BTs **52a**–**d** and **53a**–**d** to study the in vitro antitrypanosomal and antiproliferative activities (Figure 8) [112]. Specifically, the amidine group substitutions and the type of thiophene backbone impacted the biological activity. All synthesized BTs were active as both antiproliferative and antitrypanosomal agents.

The 2,2′-bithienyl-BTs with unsubstituted and 2-imidazolinyl amidine showed the most selective antitrypanosomal activity. The 2,2′-bithiophene BTs showed the most selective antiproliferative activity, whereas all 2,2′-bithienyl BTs were selectively active against lung carcinoma. The BT with an unsubstituted amidine group also produced strong antiproliferative effects. The pronounced antiproliferative activity of the BTs was attributed to different cytotoxicity mechanisms. Cell cycle analysis and DNA binding experiments provided evidence that BTs are localized in the cytoplasm and do not interact with DNA.

All amidino-substituted BTs, **52a**–**d** and **53a**–**d**, were tested in vitro for their antiproliferative activity against human cancer cell lines, including HCT116 and SW620 (colon carcinomas), H460 (lung carcinoma), MCF-7 (breast carcinoma), PC3 (prostate carcinoma) and HeLa (cervical carcinoma), and HEK 293 (human embryonic kidney) cells. The 2,2′-bithiophene-BT **53a** displayed very potent and selective activity against H460 cells, with an IC_50_ value of 0.02 μM, which was twice the magnitude of the IC_50_ values for the other cancer cells tested. All other 2,2′-bithienyl-BTs, **53b**, **53c**, and **53d**, also showed selective activity against H460 cells, with IC_50_ values of 0.2 μM, compared with doxorubicin (IC_50_ = 0.04 ± 0.01).

4NO_2_-2ATP was condensed with 4-N-substituted benzaldehydes to afford the targeting fluorescent 2-(*N*,*N*-dimethyl/diphenyl-aminophenyl)-5-substituted-BTs **54a**,**b** in 73.5% and 69.1% yields, respectively. Then, the nitro group of compounds **54a**,**b** was reduced to the amine derivatives **55a**,**b** (81.3 and 78.8%, respectively) with Pd(OAc)_2_ in ethanol and reacted with the corresponding aldehyde under stirring in methanol to afford the imines **56a**,**b** in 73.8 and 70.2% yields, respectively (Figure 7) [113]. The compounds were analyzed for their photophysical properties, absorbance, and fluorescence in different organic solvents. The photophysical properties of BTs **54a**,**b** and **56a**,**b** were strongly impacted by the solvent used. In DMSO, all BTs had higher fluorescence intensities, showing that solvent polarity has a great impact on their excited-state properties. For BT **54a**, the λmax values ranged from 337 nm to 389 nm, and BT **54b** showed λmax values ranging from 373 nm to 394 nm, with the highest absorption observed in DMSO, whereas BTs **56a** and **56b** presented λmax values ranging from 387 to 417 nm and 393 nm to 423 nm, respectively. For BT **54a**,**b**, the fluorescence emission maxima ranged from 495 to 522 nm and 512 to 531 nm, respectively, with their highest values observed in ethanol. Conversely, BTs **56a**,**b** displayed emission maxima ranging from 518 to 537 nm and 518 to 571 nm, respectively, with the highest fluorescence observed in DMSO and ethanol, respectively.

All BTs were evaluated for cytotoxicity across anticancer cell lines and inhibition against VEGFR-2 kinase using an anti-phosphotyrosine antibody test. BT **54a** showed moderate inhibitory activity with little variability (IC_50_ = 0.38 ± 0.13 μM). Conjugate **54b** had a higher potency, with an IC_50_ of 0.29 ± 0.16 μM, with lower variability. The BT **56a** showed considerable inhibitory activity, with an IC_50_ of 0.31 ± 0.08 μM, with the lowest variability, whereas BT **56b** showed an IC_50_ value of 0.23 ± 0.20 μM. Compounds **54b** and **56b** resulted in potent cytotoxic activity against MCF-7 cells (IC_50_ values of 7.06 ± 0.29 and 8.82 ± 0.37 μM, respectively), whereas compound **56b** exhibited the greatest potency in VEGFR-2 inhibition (IC_50_ of 0.23 ± 0.20 μM). Molecular docking studies indicated BT **54b** to have a stronger interaction. An ADME study showed BTs **54b** and **56b** to have enhanced lipophilicity and decreased solubility, which could influence their pharmacokinetic behavior.

2ATP was condensed with aromatic aldehydes in the presence of the prepared Co/Niacin-MOF as an economic, efficient, sustainable, and green stable catalyst under stirring at 60–70 °C for the appropriate length of time (30–60 min) to afford 2-heteroaryl-BTs **57a**–**f** in 75–95% yields (Table 20) [73]. All compounds were screened as acetylcholine esterase (AChE) inhibitors. Validation in molecular docking studies showed that BTs **57e** and **57f** gave good results in binding with acetylcholine esterase. The data from in vitro studies showed that compound **57e** had a promising value (59.8% inhibition, IC_50_ = 5.25 μg/mL) compared with the Alzheimer’s reference drug donepezil (74.89% inhibition, IC_50_ = 4.19 mg/μL).

### 2.2. Condensation with Carboxylic Acids and Their Derivatives

In a simple trituration method, 2ATP was condensed with N-protected amino acids using molecular iodine as a mild Lewis acid catalyst to synthesize the 2-substituted BTs **58a**–**f** in 66–97% yields and **59a**–**f** in 54–98% yields (Figure 9) [114]. The reaction was carried out in solvent-free conditions for 20–25 min to provide the products with moderate to excellent yields (54–98%).

Graphene oxide (GO) was used as a catalyst in the condensation of 2ATF with α-phenyl glyoxylic acids, and water was used as a solvent; the reaction was conducted at room temperature under heating conditions for 1 h to synthesize the 2-aryl-BTs **60a**–**d** (Table 21) [115]. Their neuroprotective effects were assayed in the U87 MG cell line under a H_2_O_2_-induced stressed condition and compared with the breast cancer (MCF-7) and macrophage (RAW264.7) cell lines using a cell viability assay. These 2-aryl-BTs enhanced the neuronal cell viability and protected neuronal cells from the ROS-mediated neuronal damage induced by H_2_O_2_. Furthermore, these compounds modulated catalase and enhanced the catalase activity by up to 90% during the H_2_O_2_ exposure in the U87MG cell line. Molecular modeling studies in the AutoDockTool-1.5.6. Lig Plot + program showed these compounds to have strong binding energies of −7.39, −7.52, −6.5, and −7.1 Kcal/mol, as observed by using the potent analogs **60b** and **60c** and the catalase enzyme, which indicated the presence of hydrophobic interactions in the catalytic site. Furthermore, a simulation of the ligand and catalase protein performed using DESMOND software showed further strengthened ligand and enzyme interactions. An in silico ADMET study revealed the drug-likeliness of these analogs. The BTs **60b**,**d** had potential catalase-modulating activity comparable with valproic acid as a standard drug. These results indicated the use of an in vivo animal model for possible therapy.

Substituted 2ATPs and substituted α-keto acids were reacted by a decarboxylative cross-coupling reaction under blue LED (λ = 435–445 nm) irradiation without using any photocatalyst or metal at room temperature for 8 h to afford unsubstituted and substituted BTs, **61a**–**p**, in moderate to good yields (40–88%) (Table 22) [116]. The reaction was carried out without any photocatalyst or metal. An electron donor–acceptor complex (EDA) formed with α-keto acid, and 2ATP drove the formation of the corresponding BT. The BT **61n** had the lowest yield (40%), the 5ClBT **61o** had a yield of 61%, and the 2MeBT **61p** had a yield of 53%; the 2-phBT **61a** had the highest yield (88%) yield. The 2-(*o*OHPh)BT **61e** had a yield of 73%), the 2-(*p*OMePh)BT **61j** a yield of 74%, the 2-(*p*MePhBT) **61h** a yield of 58%, the 2-(*p*ClPh)BT **61m** a yield of 77%, the 2-(*p*FPh)BT **61c** a yield of 73%, and the 5Cl,2PhBT **61b** a yield of 83%).

The 2PhBT **61a** had the higher yield (88%), whereas BT **61n** had the lowest yield (40%). Substituents in the *para* position of the phenyl ring just as the compounds **61c**,**m** with strong deactivating effect (F and Cl) with yields of 73 and 77%, and compounds **61h**,**j** with strong activating effect (OH and OMe) with yields of 73 and 77%, respectively), reflected a decrease compared to unsubstituted 2PhBT **61a**. Also, the yields of the BTs with activating groups had lower yields than those of BTs with deactivating groups. The 2-alkyl BTs **61n**–**p** had the lowest yields (40, 61, and 53%, respectively). The 4-amino substituted α-keto acid (R^3^ = *p*-aminoPh) could not be transformed to the BT **61q** in the present reaction condition, whereas from α-ketoglutaric acid, the BT **61s** was obtained in traces.

A decarboxylative coupling of α-keto acids with 2ATP was carried out in a one-pot reaction using an aqueous suspension of K_2_CO_3_ as a base under grinding at room temperature for the appropriate time for the synthesis of the 2-aryl-BTs **62a**–**h** (Table 23) [117]. The protocol was carried out with excellent tolerance of functional groups in yields of 95, 97, 93, 95, 92, 98, 95, and 91%. The advantages of this method were short reaction times, a straightforward work-up, and catalyst-free and chromatography-free purification. On comparing BTs **62e** (92%) and **62f** (98%), it was clear that an electron-withdrawing group at the meta position of the phenyl ring disfavored the yield, whereas electron-donating groups at the para position favored the yield compared with the 2PhBT **62a** (95%).

On comparing 3NO_2_PhBT **62e** (92%) with 2PhBT **62a** (95%), it was clear that an electron-withdrawing group at the meta position of the phenyl ring disfavored the yield, whereas an electron-donating in 3MePhBT **62g** (95%) favored the yield. However, the best electron-donating groups at the para position were in 4MeOPhBT **62f** (98%), the yield of which was more highly favored, surprisingly, and with electron-withdrawing groups at this position in 4FPhBT **62b** (97%) the yield also increased, maybe due to the resonance effect of the fluor atom in this case.

The Schiff base BTs **63a**–**g** were synthesized in two steps: (a) 4-minosalicylic acid and 5-aminosalicylic acid were reacted with 2ATP in polyphosphoric acid to produce the intermediates 2-(2-hydroxyaniline)BT and 2-(3-hydroxyaniline)BT. Finally, (b) these intermediates were treated with a series of trisubstituted aldehydes by stirring a few drops of acetic acid in ethanol at room temperature for 8 h to produce the **63a**–**g** BTs in good yields (85–91%) (Figure 10) [118]. These compounds were evaluated for in vitro antibacterial activities on *Bacillus sps*, *S. aureus*, *K. pneumoniae*, and *E. Coli* strains and antifungal activity on the *C. Albicans* strain using the ‘micro-broth dilution method’ (MICs in μg/μL). All the compounds displayed very good antibacterial activity for the *S*. *aureus*, *E*. *coli*, and *K. pneumonia strains* (MIC = 0.8 μg/mL), compared with ciprofloxacin (2.0 μg/mL), and antifungal activity for *C. albicans*, compared with fluconazole (16 μg/mL), as standard drugs. Molecular docking studies were performed to understand the binding mechanism. Molecular electrostatic potential (MEP) was used to evaluate the reactivity of the molecules. Antimicrobial activity was correlated with the calculated HOMO–LUMO gap, chemical hardness, and global softness. BT **63e** exhibited very good antimicrobial activity, less toxicity, and more chemical reactivity, as confirmed by a smaller HOMO–LUMO gap, a lesser electrophilicity index, a higher global softness, and a lesser chemical hardness. The evaluation of DNA cleavage of **63e** against MCF-7 breast cancer cells revealed 85.82% inhibition of cancer cells at 200 μg/μL. In addition, **63e** showed less toxicity to normal cells at the concentration required to produce an anticancer effect (IC_50_ = 973 μg/μL).

2ATP was condensed with the corresponding 2-aryl-1*H*-benzimidazole-5-carboxylic acid in the presence of polyphosphoric acid (PPA) via heating to 250 °C and 20 bar for 3 min to produce BTs **64a**–**h** in moderate yields (40–72%) (Figure 11) [119]. NIH3T3 (ATCC CRL-1658), Caco-2 (ATCC HTB-37), and A549 (ATCC CCL-185) cell lines were used to test the cytotoxic effects of the BTs. All the BTs exhibited low to moderate activity against the tested cancer cell lines. Compound **64e** had the most potent activities against A549 and Caco-2 cells, with IC_50_ values of 73.76 ± 2.54 μM and 73.15 ± 2.84 μM, respectively. This result suggested that the methyl group of the phenyl ring group may affect the antiproliferative activity to a certain extent. Also, BT **64e** displayed antiproliferative activity similar to cisplatin against NIH3T3. On the other hand, BT **64c** displayed significant potency against A549 and Caco-2 cells, with IC_50_ values of 98.83 ± 3.71 μM and 77.80 ± 3.19 μM, respectively. In addition, BT **64e** showed a similar selectivity to that of cisplatin.

Aliphatic dicarboxylic acids were condensed with 5-amidino-2ATPs in polyphosphoric acid (PPA) at 180 °C to afford a series of symmetric bis-6-amidino-BTs **64**–**71** with aliphatic central units, as previously designed, with 24–71% yields (Figure 12) [120]. The BTs were isolated as methanesulfonates by an additional acid–base reaction because these salts are more stable and soluble in water. All BTs were evaluated for their efficacy against bloodstream forms of the African trypanosome *Trypanosoma brucei*. Most of the tested compounds were more potent than fexinidazole (2.40 μM), with EC_50_ values against *T. brucei* ranging from 0.51 to 5.39 μM. With the exception of BTs **68a**–**c**, trypanocidal activity decreased in the following order: unsubstituted amidine > pyrimidine > imidazoline. The aliphatic spacer between the two BTs also influenced the activity. The bis-BTs **70a**–**c** with a cyclohexyl unit had the most pronounced impact, and bis-BT **70a**, which has an unsubstituted amidino moiety, displayed a remarkable potency (EC_50_ = 0.51 nM). BTs **71a**–**c**, which have conformationally constrained ethenyl spacers, showed better activity than the corresponding conformationally unrestricted ethyl spacers **64a**–**c**. All BTs with conformationally unrestricted alkyl chains, with the exception of the most active BT, **68c** (n = 6; EC_50_ = 0.028 ± 0.001 μM), and BTs **64**–**69**, with three to eight different lengths (n = 4 (**66a**–**c**; 0.063, 1.14, 0.19 μM)), were generally optimal. On the other hand, BT **70a** showed sub-nanomolar in vitro potency with good selectivity over mammalian cells (>26,000-fold). In all the experiments, mice treated with a dose of 20 mg kg^−1^ were cured of stage 1 trypanosomiasis. Compound **70a** displayed a favorable in vitro ADME profile, except for its low membrane permeability, while **70a** was also active at low nanomolar concentrations against cultured asexual forms of the malaria parasite *Plasmodium falciparum*. On these bases, compound **70a** was considered a lead with therapeutic potential.

2ATP was condensed with ethyl oxalate to afford the ethyl 2-carboxylateBT **72**, which suffered hydrazinolysis to be transformed to the acid hydrazide **73**, the precursor used for the preparation of the hydrazone derivatives **72**–**76** (Figure 8) [121]. BTs **74a**,**b** were obtained in 87 and 92% yields, respectively, through the interaction of **73** with either 4-chloro or 4-f luorobenzoyl chloride, respectively, in glacial acetic acid. Also, compound **73** was reacted with *p*-substituted isothiocyanates to afford the respective thiosemicarbazides **75a**–**d** in 84–90% yields. The thiosemicarbazides **75a**–**d** were refluxed in 2N NaOH to suffer cyclization into their 1,2,4-triazole-3-thiols **76a**–**d** in 31–63% yields. Also, compound **73** was reacted with 4-bromo benzenesulfonyl chloride or tosyl chloride in glacial acetic acid to afford the sulfonamides **77a**,**b** in 84 and 78% yields, respectively. Finally, acid hydrazide **73** was used as a precursor for the synthesis of the hydrazones **78a**–**g** in 88–93% yields through a reaction with aromatic and heteroaromatic aldehydes. The antitumor activities of the synthesized hydrazone BTs **73**–**76** were evaluated. Also, they were tested in vitro against colorectal carcinoma (HCT-116), hepatocellular carcinoma (HepG2), prostate cancer (PC-3), mammary gland cancer (MCF-7), and epithelioid carcinoma (HeLa). The most active compounds were **78e**, **78f**, **75b**, **76c**, and **77b**, exhibiting IC_50_ values comparable to that of lapatinib, the reference drug. The most potent compounds for EGFR inhibitory activity were **78e** and **78f**, with IC_50_ values of 24.58 and 30.42 nM, respectively, compared to the standard drug lapatinib (17.38 nM). Molecular modeling studies were conducted, including docking into the EGFR active site and surface mapping for all compounds. The hydrazones **78e** and **78f** showed superior binding with EGFR, showing them to be excellent candidates for targeted antitumor therapy through EGFR kinase inhibition.

The condensation reaction of 2ATP with acyl chlorides or acid anhydrides was performed in the presence of KF-Al_2_O_3_ as a heterogeneous base catalyst under stirring with acetonitrile at room temperature for 30 min to afford 2-substituted BTs **79a**–**k** in 87–97% yields (Table 24). The catalyst showed good recyclability [122]. It was observed that electron-withdrawing functionalities gave higher isolated yields (**79a**,**b**; 97 and 95%). On the other hand, electron-donating groups (**79d**–**f**; 94, 91, 90%) gave lower isolated yields.

2ATP was condensed with nine pyrrolidinone ester derivatives under microwave irradiation in the presence of a heterogeneous eutectic mixture of (CuCl_2_ or NiCl_2_/urea) and Cu- or Ni-doped TiO_2_ nanoparticles to produce N-aryl pyrrolidinone-BTs **80a**–**i** in 83–96 and 78–91% yields, respectively (Figure 13) [123]. Eco-friendliness and short reaction times (5 min) were the advantages of this protocol. All synthesized BTs were screened for their antibacterial activity against Gram-positive and Gram-negative bacterial strains: compounds **80a** and **80b** were active against *E. coli* and* P. aeruginosa*, with an MIC of 12.5 µg/mL. Compounds **80c**, **80e,** and **80f** showed important activity against *MRSA* and *BS*, with MIC values ranging from 12.5 µg/mL to 25 µg/mL. Also, compounds **80c** and **80f** were active against the four tested bacteria strains, with MIC values of 12.5 for *EC*, 12.5 for *PA*, 12.5 for *BS*, and 18.75 μg/mL for *MRSA*, compared with the antibiotic ciprofloxacin as a standard control.

The *p*-nitro benzoyl chloride was condensed with 5-methoxy-2ATP in toluene as a dissolvent upon heating for 1 h to afford the BT derivative **81**, which was reacted in ethanol with stannous chloride upon heating to 80 °C for 3 h; then, paraformaldehyde dissolved in fresh NaOMe in MeOH solution was mixed and stirred at room temperature for 4 h. After that, sodium boron hydride was added, and the solution was refluxed for 1.5 h to afford BT **82** in a 33% yield. A solution of **82** and ICl in acetic acid was refluxed upon stirring overnight for 10 h to afford BT **83a** in a 35% yield, which was cooled to −78 °C and reacted with BBr_3_ upon stirring at room temperature for 16 h to afford **83b** in a 37% yield. The neopentylboronic ester **84a** could be quantitatively formed from **83a**, but the isolation of **84a** without hydrolysis was challenging. Partial and full hydrolysis of **84a** during isolation efforts led to unsatisfactory mixtures of **84a** alongside boronic acid **85a** and protodeborylated compound **82**. The synthesis of BT derivatives with [^211^At]3′-At-PIB-OMe **86a**,**b** was carried out using a Cu-catalyzed astatination protocol via boronic acid precursors **85a**,**b** (Figure 9) [74]. Compound **86a** had a radiochemical yield of 55% and was stable for at least 3 h in phosphate-buffered saline. Studies were carried out to evaluate the binding affinities of **86a** and **86b** against Aβ(1–40) AD plaques and their in vivo stability, biodistribution, and AD-associated plaque clearance abilities.

A series of substituted 2ATPs were condensed with a variety of aryl/alkyl nitriles under mild reaction conditions (oil bath, 70 °C) using ZnO nanoparticles as a catalyst in solvent-free conditions to produce two series of 2-substituted BTs, **87a**–**z** and **88a**–**d**, in 88–96% yields (Table 25) [124]. This method was compatible with several functional groups and exhibited excellent catalyst recyclability, easy recovery of materials, and high economy; there is no need to purify the products by recrystallization or column chromatography, and it is an environmentally benign process.

Five 2-[3-(aryl)prop-2-enenitrile]BTs **90a**–**e** were synthesized in 84–96% (4–8 min) yields by a microwave irradiation method from 2-cyano-methylBT **89** in EtOH/AcOH, obtained in a 92% yield from the condensation of 2ATP and malononitrile under microwave irradiation at 40 °C for 10 min and used as an intermediate in the synthesis of 7-amino-6-(BT-2-yl)-5-(aryl)-2-thioxo-2,3dihydropyrido[2,3-d]pyrimidin-4(1H)-ones **91a**–**e** obtained in 84–92% yields (25 min) and 1-amino-2-(aryl)pyrrolo[2,1-b][1,3]BT-3-carbonitrile derivatives **92a**–**e** obtained in 78–90% yields (20–25 min) (Figure 10) [125]. These compounds were tested as antioxidant, antimicrobial, and anticancer agents.

It was found that pyrimidine-BTs **91a**,**d** were more potent against bacteria, with MICs from 4 to 12 μmol L^–1^, than cefotaxime (MIC = 6–12 μmolL^–1^); pyrimidine-BTs **91bc**,**e** showed good activity against all the bacterial strains. This high efficacy was attributed to the presence of a BT and thiophene moieties, as in pyrimidine-BT **91a**, and to the presence of an electron-withdrawing (fluoro) group in the *para* position of the phenyl ring attached to the pyridopyrimidine, as in pyrimidine-BT **91d**. Pyrimidine-BT **91a** had equipotent activity with MICs of 6 and 12 μmol L^−1^ similar to those of cefotoxime against *Bacillus subtilis* and *Chlamydia pneumonia*, respectively, whereas **91e** was equipotent (MIC = 8 μmol L^−1^) with respect to cefatoxime against *Salmonella typhi*. Pyrrolo-BTs **92a**–**e** effectively inhibited the growth of the tested bacteria strains. PyrroloBT **92a** was more potent against *Staphylococcus aureus* and had equipotent efficacy against *Bacillus subtilis* and *Chlamydia pneumonia* compared with cefotaxime; this was attributed to the presence of a thiophene moiety. In addition, pyrrolo-BT **92d** (MICs from 4 to 10 μmol L^–1^) had extraordinary activity toward all bacteria due to the presence of a *p*-fluorophenyl substituent which increased the antibacterial potency compared to the reference drug. Also, pyrrolo-BTs **92b**,**c**,**e** exhibited good inhibition activity against all bacterial strains. The cytotoxic activity of pyrimidine-BTs **91a**–**d** and pyrrolo-BTs **92a**–**d** against the colon cancer HCT-116, lung cancer NCI-H460, and liver cancer HepG2 human cell lines was observed and compared with that of doxorubicin as a reference drug. The series of pyrimidine-BTs **91a**–**e** exhibited higher cytotoxic activity than the pyrrolo-BTs **92a**–**e**. Pyrrolo-BT **91a** containing thiophene (IC_50_s = 0.66, 0.45, and 0.70 μmol L^−1^), pyrrolo-BT **91d** containing *p*-fluorophenyl (IC_50_s = 0.28, 0.39, and 0.14 μmol L^−1^), pyrrolo-BT **92a** with thiophenyl (IC_50_s = 1.10, 0.49 and, 0.89 μmol L^−1^), and pyrrolo-BT **92d** containing *p*-fluorophenyl (IC_50_s = 0.45, 0.47, and 0.59 μmol L^−1^) were found to be more potent and efficacious than doxorubicin as the reference drug (IC_50_s = 1.98, 0.52, and 1.12 μmol L^−1^). It was found that the presence of the *p*-fluorophenyl group improved the antitumor activity more than thiophene. In addition, pyrrolo-BT **91b** (IC_50_s = 2.56, 2.89, and 2.68), and pyrrolo-BT **91e** (2.60, 3.00, and 3.08 μmol L^−1^) exhibited good cytotoxic effects towards the three tumor cell lines. On the other hand, 5-methylfurany **92b** (IC_50_s = 3.90, 4.08, and 4.50 μmol L^−1^) showed good antitumor activity against all tumor cell lines. Compounds with 5-naphthalenyl side chains in both pyrimidine-BT **91c** and pyrrolo-BT **92c** showed moderate antitumor activity. From the above structure–activity relationships (SARs), the most selective pyrimidine BT and pyrrolo-BT on the studied cancer cell lines bore *p*-fluorophenyl and thiophene moieties. Thus, pyrimidine-BTs **91a**,**d** and pyrrolo-BTs **92a**,**d** were considered promising candidates as pharmacophores against cancer cell lines. A series of pyrimidine-BTs **91a**–**e** and pyrrolo-BTs **91a**–**e** were tested for antioxidant activity, as reflected in the ability to inhibit lipid peroxidation in rat brain and kidney homogenates using (3 ethylbenzthiazoline-6-sulfonicacid) an ABTS free radical scavenging assay. Pyrimidine-BTs **91a**,**d** (91.2 and 92.8% yields, respectively) and pyrrolo-BT **92d** (90.4%) showed inhibition levels higher than that of Trolox (89.5%), while pyrrolo-BT **92a** (88.7%) showed nearly equipotent inhibition activity.

**Scheme 10 ijms-26-05901-sch010:**
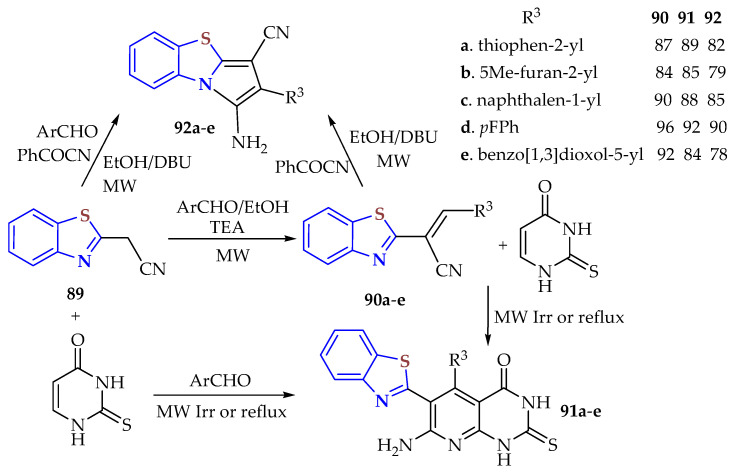
Condensation of 2ATP with malononitrile to afford 2-[3-(aryl)prop-2-enenitrile]BTs **90a**–**e** as intermediates in synthesizing BT derivatives **91**,**92a**–**e** by the MWI method in % yields.

The nano-catalyst aminopropyl-1,3,5-triazine-2,4-diphosphonium tetrachloroferrate immobilized on halloysite nanotubes [(APTDP)(FeCl_4_)_2_@HNTs] was prepared to be used as a highly effective agent in the condensation reaction of 2ATP with benzonitriles containing various electron-withdrawing and -donating substituents for the synthesis of several BTs, **93a**–**o**, in excellent yields (90–96%). The reaction was performed in the absence of a catalyst under solvent-free conditions at 80 °C within 4–5 h (Table 26) [126]. Additionally, the use of dinitriles such as 1,3-dicyanobenzene and 1,3-phenylenediacetonitrile, using a 1:1 molar ratio, gave the corresponding mono-benzonitrile-BT **9**3t**** and mono-methylphenylacetonitrile-BT **9**3u** in 90 and 85% yields, respectively,** using a 1:1 molar ratio. Also, bis- and tris-BTs **93v**–**x** were smoothly synthesized from dinitrile and trinitrile in those conditions in 78 and 75% yields. The advantages of this protocol were its easy work-up, high purity, high catalytic activity, reusability, and the easy recovery of the catalyst. On the other hand, the competitive condensation of 2ATP with an equimolar mixture of an aromatic nitrile (4-methylbenzonitrile) and an aliphatic nitrile (heptanenitrile) was examined under the same conditions. The results showed that 4-methylbenzonitrile was transformed to the corresponding BT **9**3j**** (90%) in the presence of heptanenitrile. This outcome indicated the selectivity performance of the catalyst.

### 2.3. Condensation with Ketones

2ATP was condensed with several 7-benzylidenebicyclo[3.2.0]hept-2-en-6-ones to afford a series of thirteen 2-[2-(2-phenylethenyl)cyclopent-3-en-1-yl]-BTs, **94a**–**m**, in 93–98% yields (Figure 14) [127]. The antiproliferative activities of all compounds were tested against C6 (rat brain tumor) and HeLa (human cervical carcinoma) cell lines using BrdU cell proliferation ELISA assays, with the use of 5-fluorouracil (5-FU) and cisplatin as standard drugs. Compound **94e** was the most active against C6 cell lines, with an IC_50_ = 5.89 μm (5-FU, IC_50_ = 76.74 μm, and cisplatin, IC_50_ = 14.46 μm). But compound **94c** was the most active against HeLa cell lines, with an IC_50_ = 3.98 μm (5-FU, IC_50_ = 46.32 μm, and cisplatin, IC_50_ = 37.95 μm). On the other hand, computational studies of all compounds using the B3LYP/6-31G+(d,p) level in the gas phase were performed. The calculated results were compatible with the experimental IR and NMR data. Molecular electrostatic potential (MEP) maps allowed the identification of the region in **94c** that is biologically active against HeLa and that in compound **94e** which is active against C6. Molecular docking analysis was used to determine the biological activities of these molecules. The appropriate target proteins (PDB codes: 1JQH for the C6 cells and 1M17 for the HeLa cells) were used for **94c** and **94e**, demonstrating high antiproliferative activity against both cell lines. These compounds were proposed as promising candidates for anticancer drugs.

### 2.4. Miscellaneous Condensations

2ATP and both electron-rich and electron-poor substituted styrenes were condensed in a one-pot methodology for the synthesis of the pharmacologically active 2-arylBTs **95a**–**f** in 30–66% yields (Table 27). The reaction was carried out with Pd(OAc)_2_ in AcOH as a solvent and heating at 90 °C for 12 h in the presence of O_2_. The process involved sequential C-C/C-N bond cleavage followed by C-N/C-S bond formation [128]. To further the scope of the reaction, 2ATP was reacted with styrene under standard conditions. However, no desired product was obtained, and the starting materials were isolated.

2ATP disulfide was reacted with terminal alkynes under aerobic oxidative C(sp)–S coupling in a visible-light-driven copper-catalyzed reaction (Table 28) [129]. The photochemical reaction was achieved using a wide range of thiol dimers and alkynes with good chemoselectivity and excellent conversion. The utility of alkynyl sulfide formations as isolated intermediates (35–92%) was demonstrated in the construction of the 2-phenylBTs **96a**–**n** in 55–79**%** yields via “thia-Wolff rearrangement” in the presence of AgNO_3_, using visible light with 9-mesityl-10-methylacridinium ions (Acr^+^–Mes) as a photoredox catalyst system. BTs with electron-donating groups on the benzene ring increased the yield, while BTs **96b**–**d** and BTs with electron-withdrawing groups at the ortho (**96f**), meta (**96k**), or para (**96e**,**g**,**h**) positions decreased the yields compared with the 2PhBT **96a**. The BT with a pyridinyl ring instead of a benzene ring in **96l** (59%) decreased the yield. Curiously, napthalenyl and phenanthrenyl groups, instead of the phenyl ring, resulted in high yields of **96m**,**n**. This reaction failed with aliphatic terminal alkynes.

In a metal-free and environmentally friendly approach, 2ATP was condensed with methyl arenes under visible-light irradiation using eosin Y as a photocatalyst, ethanol–water (1:2) as a green solvent at room temperature, and atmospheric air as an oxidant to synthesize the biologically important BTs **97a**–**q** (Table 29) [130]. This methodology demonstrated a broad substrate scope, yielding the desired products in good to excellent yields (83–92%). The advantages of this procedure included its environmental friendliness, low cost, non-toxicity, green solvent, ease of handling, and use of visible light as a renewable energy source. All the BTs were obtained in good yields when the substituent was in the meta and ortho positions (**97b**–**h**,**j**; 85–92%). When the methyl arene had F, Cl, and Br (**97b**–**d**; 91, 90, and 90%, respectively) and OMe (**97j**, 86%) in the *p*-position, the yield obtained with electron-withdrawing groups was higher than that obtained with electron-donating groups. Additionally, methyl arene with multiple substituents yielded 85% (**97i**). The use of heteroaromatic methyl arenes (**97n**–**q**,**v**,**w**) afforded BTs with good yields (83–84%).

2ATP was condensed by an oxidative coupling with substituted aryl methyl amines in the presence of a Ba-doped CoMoO_4_ system as a catalyst to synthesize several 2-aryl BTs **98a**–**k** under visible-light irradiation (Table 30) [131]. The reaction was carried out under atmospheric air and had good to excellent yields (78–98%). The advantages of this protocol were its atom economy, functional group tolerance, a wide range of substrate scopes, and suitability for scale-up reusability. Aryl methyl amines with an OMe as an electron-donating group at the para position of the benzene ring favored the yield, as in **98b**, but an *^i^*Pr group (**98d**) decreased the yield due to steric effects. On the other hand, electron-withdrawing groups, such as Cl or F (**98h** and **981**), decreased the yield, compared with 2PhBT **98a**. The aryl methyl amines with electron-donating or electron-withdrawing substituents at the para position, BTs **98b**,**d**,**h**,**i**, were obtained with good to excellent yields. Electron-donating or electron-withdrawing substituents at the ortho position of the benzene ring decreased the yields (**98e**,**j**), but electron-withdrawing substituents decreased the yields more. The OPh group at the meta opposition of the phenyl ring produced only a little decrease (**98f**).

The condensation of 2ATP with aromatic primary alcohols via the acceptorless dehydrogenative coupling (ADC) method was carried out in the presence of Pd(II)N^N^S as a pincer-type catalyst for the synthesis of the pharmaceutically important BT derivatives **98a**–**u** (Table 31) [132]. Several primary aromatic alcohols, such as electron-releasing and -withdrawing alcohols, sterically hindered alcohols, heterocyclic alcohols, polycyclic alcohols, and aliphatic alcohols, were coupled with 2ATP. *p*-substituted benzyl alcohols, including electron-donating 4-methyl and 4-isopropyl benzyl alcohol, were coupled with 2ATP to afford the corresponding BTs **99a**,**b** in 90 and 86% yields, respectively. Interestingly, alcohols containing electron-poor groups, such as 3-fluro, 4-choloro, and 4-nitro benzyl alcohols, were well-tolerated to obtain BTs **99g**–**i** in 69, 72, and 53% yields, respectively. Benzylic alcohols with sterically hindered 2-substituted alcohols, such as 2-Br, 2,5-Cl_2_, 2-amino, and 2-NO_2_-benzyl alcohols, gave BTs **99j**–**m** with 58, 63, 71, and 47% yields, respectively. Next, benzyl alcohols containing electron-rich groups like 2,3-dimethoxy and 2,5-substituted groups were reacted to afford BTs **99n**,**o** in 74 and 71% yields, respectively. Reactions with heteroaromatic benzyl alcohols such as 2-pyridine methanol, piperonyl alcohol, and thiophene-2-methanol proceeded smoothly to furnish BTs **99p**–**r** in 69, 73, and 72% yields, respectively. Interestingly, the reaction of cinnamyl alcohol and polycyclic alcohols, such as 4-biphenyl benzyl acohol, 1-naphthalene methanol, and pyrene 2-methanol, afforded the corresponding BTs **99s**–**v** in 66, 79, 75, and 56% yields, respectively. The use of aliphatic alcohols such as propan-1-ol, 2-methylpropan-1-ol, and hexan-1-ol gave the BTs **99w**–**y** in 52, 53, and 48% yields, respectively. To understand the electronic effect, an intermolecular competitive experiment involving 4-methyl benzyl alcohol and 4-chloro benzyl alcohol was carried out under the same reaction conditions. It was found that the electron-rich benzyl alcohol is more reactive to give BT **99b** with a 53% yield than the electron-deficient benzyl alcohol to give BT **99h** with a 32% yield.

The BTs were synthesized via C−S and C−N bond formations, yielding up to 93%. This method was sustainable, with excellent yields, 1 mol% catalyst loading, inexpensive primary alcohol, and only water and hydrogen gas as by-products. A mechanism was proposed via in situ aldehyde formation and dehydrogenation of primary alcohols. The utility of this catalytic procedure was illustrated by the large-scale synthesis of the 2-(4-methoxyphenyl)benzo[d]thiazole **99a**.

The treatment of the respective substituted (2-isocyano-phenyl)(alkyl)sulfanes, derived from 2ATP, with a series of diarylphosphine oxides under irradiation with 23W white LED light for 13 h at room temperature in an air atmosphere, using the photocatalyst Rose Bengal and DMF as a solvent, gave the 2-phosphorylBTs **100a**–**l** in 65–89% yields (Figure 15) [133]. On the other hand, electron-poor and electron-rich substituents on the benzene ring in the 2-isocyanoaryl thioethers produced the substituted 2-phosphorylBTs **100m**–**t** in 64–88% yields.

Several H-phosphorus oxides were well-tolerated, regardless of whether the ring of diarylphosphine oxides was substituted with either electron-deficient, electron-rich, or steric-hindered groups; the reactions proceeded to afford BTs **100a**–**i** in medium to good yields (65–87%). Additionally, dimethyl-substituted H-phosphorus oxide was used to give BT **100j** in a moderate yield (58%). The same procedure was used with piperonyl and naphthyl phosphine oxide derivatives, affording the BTs **100k**,**l** (79 and 81% yields, respectively). On the other hand, functional groups with electron-rich (OMe and Me) and electron-poor (F, Cl, Br, and –SO_2_Me) substituents on the BT ring were also tolerated to afford BTs **100m**–**t** in good yields (64–88%).

A metal-free radical cascade cyclization of substituted 2-isocyanophenyl-methyl-sulfanes with 5.0 mL of the corresponding primary or secondary alcohol in the presence of the radical initiator di-*tert*-butyl-peroxide (DTBP) was refluxed for 6 h at 120 °C (Figure 16) [134]. The desired 2-hydroxyalkylBTs **101a**–**i** and **102a**–**v** were obtained in yields ranging from 34 to 92%. The advantages of this reaction were its mild reaction conditions, its being a metal-free reaction, the good functional group tolerance, and the broad substrate scope.

This cascade cyclization reaction of 2-isocyanoaryl thioethers proceeded smoothly with alcohols such as isopropanol, ethanol, and 2-butyl alcohol to give the desired BTs **101a**,**c**,**f** in good yields (68.92%). Other alcohols such as methanol, *n*-propanol, *n*-butyl alcohol, cyclopentanol, and cyclohexanol afforded BTs **101b**,**d**,**e**,**g**,**h** with moderate yields (51–68%). Additionally, the benzyl alcohol gave BT **101i** in a 34% yield. On the other hand, several 5-substituted 2-isocyanoaryl thioethers reacted well with isopropanol to yield BTs **102a**–**g** in 57–91% yields. It was found that substituents at the C5 position of 2-isocyanoaryl thioethers had little effect on the reaction. Other substituents such as phenyl, 4-chlorophenyl, 4-methylphenyl, naphthalen-2-yl, and phenanthren-9-yl also proceeded smoothly to synthesize BTs **102h**–**j** and **102r**,**s** in good yields (53–86% yields). Substituents at the C5 position were also well-tolerated to afford BTs **102k**–**m** in 76–82% yields. C4 substituents were also tolerated to afford BT **102o** in a 73% yield. Moreover, it was found that heteroaryl substituents also gave BTs **102t**–**v** in 64–84% yields. 3-chloro, 4,6-dichloro, and 5,6-dimethyl substituents also reacted with isopropanol to give BTs **102p**,**q** in 68 and 84% yields, respectively.

2-Isocyanoaryl thioethers derived from 2ATPs were cyclized with ethers in an oxidant-free and metal-free visible-light-induced reaction to access the ether-functionalized BTs **103a**–**v** (Figure 17) [135]. The cyclization reaction was carried out with 1,2,3,5-tetrakis-(carbazol-9-yl)-4,6-dicyanobenzene (4CzIPN) as a photocatalyst and the irradiation of blue LED light in a nitrogen atmosphere in mild reaction conditions, and the procedure was characterized by operational simplicity. This strategy provides an efficient approach to access various ether-containing BTs in acceptable to good yields.

Both electron-rich groups (−Me and −OMe) and electron-poor groups (−F, −Cl, and −Br) were well-tolerated in the reactions and smoothly afforded BTs **103a**–**k** in moderate to good yields (42–70 %). Notably, the Br substituent at both the 4- and 5-positions of the phenyl rings gave BTs **103e**,**g**,**l** in relatively low yields (64, 43, and 47% yields, respectively). Moreover, this methodology was extended to 1,3-dioxolane and several substituted isocyanides to afford BTs **103m/m′**-**p/p′** in 65 (2:1), 61 (2:1), 57 (3:1), and 60% (2:1) yields, respectively. It was observed that the BTs showed a preference for C2 selectivity over the C4 position. The synthesized compounds combine BT with an ether functional group representing key structural motifs of various biologically active molecules of interest in medicinal chemistry.

A photo-induced cascade sulfone alkylation/cyclization of 2-isocyanoaryl thioethers and α-iodosulfones was carried out to afford alkyl/benzyl-sulphonyl BTs in up to 97% yields **104a**–**zf** (Figure 18) [136]. This visible-light-triggered reaction not only occurs under extremely mild reaction conditions but also does not require the presence of a photosensitizer. The photocatalytic process is triggered by the photochemical activity of in situ-generated electron donor–acceptor complexes arising from the association of 2-isocyanoaryl thioethers and α-iodosulfones. The radical pathway was confirmed by UV–vis spectroscopy, radical trapping, a Job plot, and on/off irradiation experiments.

A visible-light-induced radical addition/cyclization cascade reaction of 2-isocyanoaryl thioethers was achieved under metal-free and mild conditions (Table 32) [137]. This method utilizes hydrazines as radical precursors and efficiently accesses a series of 2-aryl and 2-alkyl BTs, **105a**–**r** and **106a**–**k**, in good yields (71–83% and 70–81%).

In the reaction of 2-isocyanoaryl thioethers, the groups on the benzene ring, electron-donating (Me and MeO) and electron-withdrawing groups (halogens, CF_3_, CN, and NO_2_), were compatible with the process to afford BTs **105b**–**k** in good yields (71–83% yields). No influence of the position of the substituents was observed on the yields. In addition, the disubstituted substrate reacted well to afford BT **105l** in a 79% yield. Instead of aryl rings, β-naphthyl-, 4-pyridyl-, and 2-furanyl-substituted hydrazines were used in this method to afford BTs **105m**–**o** in 74–80% yields. It is worth mentioning that benzylhydrazine, isopropylhydrazine, and cyclopropylhydrazine could furnish BTs **105p**–**r** in good yields (70–75%) when a series of substituted (2-isocyanophenyl)(methyl)sulfanes were used. In general, substrates bearing electron-rich (Me, Et, and MeO) or electron-deficient groups (halogens, CF_3_, and CN) were tolerable and afforded BTs **106a**–**j** in good yields (71–82% yields). When a disubstituted substrate was used, BT **106k** was formed in a 73% yield.

The 2-guanidinoBT **107**, obtained from condensation of 2ATP with cyanoguanidine, was used for the development of a range of synthetic methods with various reagents, such as α,β-unsaturated carbonyl, 2-cyano-three-(dimethylamino)-*N*-acrylamide, β-diketones, β-keto esters, and (*S*,*S*)-ketene dithioacetals, to afford pyrimidine-based BTs **108**–**115** in 66–72% yields as novel anticancer agents (Figure 11) [138].

The docking analysis revealed that the most active BTs, **7b**, **7c**, **13b**, **13c**, **15c**, **17d**, **18,** and **24**, fit inside the active site within the protein tyrosine kinase (PTK). It was observed that these compounds formed a hydrogen donor bond with Met318, except for **15c**. Among these compounds, BTs **15c**, **17d**, **18**, and **24** resulted in binding energies closer to that of the cocrystallized ligand 1N1, with values of −7.729, −7.321, −7.681, and −7.5790 kcal mol^−1^, respectively. Compound **13c** had a binding energy of −6.963 kcal mol^−1^, but it had four more interactions with the active site, including one arene–H interaction with Leu248 and three H-bond donors with Met318 and Thr319. Furthermore, the in silico studies and ADMET properties of the most potent BTs suggested them as promising candidates for further development, with favorable bioavailability and pharmacokinetic profiles. In vitro cytotoxicity studies were carried out against HepG2, HCT116, and MCF7 human tumor cell lines at a concentration 100 μmol mL^−1^ using the standard 3-(4,5-dimethylthiazol-2-yl)-2,5-diphenyltetrazolium bromide (MTT) bioassay. BTs **108c**, **109b**, **111d**, and **112** showed strong efficacy against the HepG2 cell line, exhibiting cell viability percentages of 61.29, 68.18, 61.04, and 66.85, respectively, compared with the standard drug (64.41%). BTs **108b**, **109a**, **110c**, and **111b** exhibited moderate activities against the same cell line, with cell viabilities of 72.70, 72.35, 72.69, and 76.02%. In addition, BTs **108b**, **109c**, and **110c** showed slightly strong efficacy against the HCT116 cell line, with viabilities of 70.62, 67.11, and 65.68%, respectively, in contrast to the standard drug with a cell viability percentage of 55.96. Furthermore, the singular BT **24** exhibited efficacy against the MCF7 cell line, with a cell viability of 69.98%, compared with the standard drug (62.76%). From the IC_50_ results, it was found that the pyrimidine-based BT **111d (IC_50_ =** 0.41 ± 0.01 μmol mL^−1^) was the most potent of the tested BTs against HepG2. Also, BT **112** (IC_50_ = 0.53 ± 0.05 μmol mL^−1^) was the second most potent compound against HepG2, followed by compound **109b** (IC_50_ of 0.56 ± 0.03 μmol mL^−1^). Notably, BTs **111d**, **112,** and **109b** showed higher potency, based on their IC_50_ data, compared with 5-fluorouracil (IC_50_ of 1.03 μmol mL^−1^) as the reference drug. Surprisingly, BT **110c** (IC_50_ of 0.02 ± 0.001 μmol mL^−1^) showed higher efficacy against HCT116 compared with 5-fluorouracil (IC_50_ of 9 ± 1.7 μmol mL^−1^). BT **110c** not only showed heightened potency related to 5-fluorouracil but also showed notable efficacy together with BTs **108b** and **109c** (IC_50_ values of 2.95 ± 0.26 and 1.033 ± 0.06, respectively). In addition, BT **115** showed an IC_50_ value of 1.485 ± 0.15 μmol mL^−1^, lower than that of the 5-fluorouracil (IC_50_ of 7.12 μmol mL^−1^), against MCF7. These results suggest that the investigated BTs exhibit potential as anticancer agents. The findings revealed that the inclusion of halogen groups, such as F and Cl, on the aryl group bonded with pyrimidine-based BTs **108a–d** resulted in an increase in their activity. In addition, the incorporation of CO_2_Et, as in **110c**, enhanced the potency in comparison to its analogs, **110a**,**b**, containing COCH_3_ and COPh, respectively. In the context of pyrimidine-based BTs **111a–d**, the presence of a OMe group amplifies the compound’s potency more significantly than those possessing halogen substituents. Notably, BTs containing SCH_3_ exhibited the lowest activity levels across the three tested cell lines.

## 3. Conclusions

In this investigation, we found that BT derivatives are compounds that have currently boomed in the last five years. In this period, at least 21 review articles appeared in the literature, and about 53 articles were found to be related to the condensation reaction of 2ATPs for the synthesis of this kind of compounds, studied for their biological activity, most of them as anticancer agents. In addition, we found that medicinal chemists also explore BT derivatives in searching for anti-Parkinson’s and anti-Alzheimer’s candidates. In some articles, the structure–activity relationships providing a framework for drug discovery and design were discussed. On the other hand, BT derivatives were explored as molecular entities endowed with two or more biological actions, called multifunctional drugs.

The most popular method to synthesize 2-substituted BTs was found to be the condensation of 2ATPs with carbonyl compounds such as aldehydes and carboxylic acids and their derivatives. The majority of these approaches have several drawbacks, such as harsh reaction conditions, high reaction temperatures, multistep processes, the need for an excessive amount of reagents, prolonged reaction times, and the employment of expensive, air-sensitive catalysts, among others. One of the main protocols for preparing BTs documented in the literature is the use of polyphosphoric acid (PPA). This popular method requires heating to high temperatures (110–220 °C) and long reaction times (1–24 h). These harsh conditions depend on the starting material’s stability, which limits the generalization of this condensation. Therefore, as an alternative to PPA, other catalyst-dehydrating agents have been designed to be used in this condensation, such as P_2_O_5_/MeSO_3_H, trimethylsilyl polyphosphate ester (PPSE), triphenylphosphine (PPh_3_), tetrabutylammonium bromide (TBAB), Samarium (III) triflate, molecular iodine, *N*-methyl-2-pyrrolidone (NMP), AlMe_3_, KF/Al_2_O_3_, CuCl_2_/K_2_CO_3_, *o*-benzene disulfonimide, acetic acid, pyridine, glycerol, DMSO, DMF, etc. These acids were effective for a wide range of aliphatic and aromatic carboxylic acids and derivatives to afford BTs in less exacting conditions and shorter reaction times. However, there is still a need to create a straightforward, mild, highly efficient, and environmentally benign method for the synthesis of BTs without the use of hazardous chemicals or reagents. Sometimes such condensation reactions have been carried out via direct heating or refluxing in solvents from high to low boiling points, improving yields. Due to their low production costs and simplicity of usage, applications of solvent-free synthetic methods to synthesize pharmacologically relevant BT derivatives have grown in favor. In this sense, the use of ionic liquids (ILs) as green solvents in these condensation reactions has gained considerably in importance due to their solvating ability, negligible vapor pressure, and easy recyclability. In addition, they have been shown to promote and catalyze these transformations due to their high polarity. ILs can also be recovered and recycled many times with marginal loss. The use of ILs such as 1-Butil-3-metil imidazolium [BMIM][BF_4_] or [BMIM][PF_6_] in this type of condensation reaction has been found to increase yields. Another method involving condensation uses microwave irradiation (MI). This method is carried out in short reaction times (5 to 60 min) with good to excellent yields (70–90%). As can be seen in the Table 33, recently, novel catalyst systems have been developed, such as homogeneous and heterogeneous catalysts, nano-catalysts, biocatalysts, photocatalysts, and combinations of these. Several examples of these systems have been proved to be excellent catalysts that reduce reaction times and increase yields with very good to excellent results (80–99%). Additionally, these methods are mild procedures, environmentally benign with good recyclability of the catalysts, and effective for a wide range of aliphatic and aromatic carbonyl compounds.

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
