# Peer review of "Condensation Reactions of 2-Aminothiophenoles to Afford 2-Substituted Benzothiazoles of Biological Interest: A Review (2020–2024)"

_ijms, 2025, doi:10.3390/ijms26125901_

Round 1

Reviewer 1 Report

Comments and Suggestions for Authors

General Comments:

This manuscript by Padilla-Martínez et al. offers a review of the recent literature (2020-2024) focused on synthesizing 2-substituted benzothiazoles (BTs) via condensation reactions of 2-aminothiophenol (2ATP). The review encompasses reactions with various partners, including aldehydes, carboxylic acids and their derivatives, ketones, and other substrates, frequently noting the reported biological activities of the resulting compounds.

Compiling recent synthetic advances in the important area of benzothiazoles is a valuable effort. However, the current manuscript presents several significant concerns regarding its analytical depth and presentation quality, which, in my assessment, preclude its publication in the International Journal of Molecular Sciences at this time. The specific points justifying this recommendation are detailed below.

Major Concerns:

  • Predominantly Descriptive Approach and Lack of Critical Analysis:

A primary concern is the largely descriptive nature of the review. Synthetic methods are generally presented sequentially (Scheme -> Conditions -> Yields -> Reported Biological Activity) without providing sufficient comparative discussion or in-depth critical analysis.

While a wide range of methods, catalysts (photocatalysts, metal catalysts, organocatalysts), and conditions (MW, US, solvent-free) are included, the manuscript would be significantly strengthened by a more meaningful comparison of their relative efficiencies, substrate scopes, inherent limitations, environmental impact ("greenness"), and potential cost-effectiveness.

The discussion of substituent effects on reaction outcomes or biological activity often appears limited to summarizing the findings of the primary source, rather than synthesizing broader trends or providing deeper rationalization across multiple studies.

As currently presented, this descriptive style makes it challenging for the reader to effectively discern the specific advantages of certain methods or to select appropriate strategies for particular synthetic targets or substrate types. The review, therefore, offers limited critical guidance beyond the compilation itself.

  • Abstract Content and Specificity:

The Abstract provides a general overview of the importance of BTs and their activities but could be more specific regarding the actual content of the review. It would benefit from clearly outlining the main types of condensation reactions covered within the manuscript.

In its current form, it focuses heavily on justifying the topic's relevance rather than serving as a concise summary of the review's scope, key findings, or trends discussed, which would be more helpful to the reader.

  • Quality of Schemes and Data Presentation:

The clarity of several schemes could be improved. In some instances, substituent labels (R¹, R², etc.) are small, cluttered, or poorly defined, making it challenging to quickly interpret the structural variations (e.g., Schemes 9, 11, 19, 39, 52, 56 present particular challenges due to density or complexity). Clear and easily interpretable graphics are essential for a high-standard publication.

Attention to detail in data presentation also requires improvement. For example, there appears to be an error in Scheme 12, where the reaction conditions header seems to be incorrectly duplicated from Scheme 11 ("Laccase/DDQ, air, Solvent, Ph 5, rt"), which is inconsistent with the described ZnClâ‚‚/HAp catalysis. Accuracy in such details is crucial for reader confidence.

The organization of complex schemes depicting multiple steps or components (e.g., Schemes 19, 39, 52, and 56) could be restructured for enhanced clarity and readability.

Conclusion:

Due to the concerns outlined above regarding the predominantly descriptive approach lacking sufficient critical analysis, issues with the quality of presentation (Abstract, schemes, data accuracy), the manuscript in its current form does not fully meet the standards expected for publication in the International Journal of Molecular Sciences. While the topic is relevant, the review's impact and contribution, beyond the literature compilation itself, appear limited in the current format. For these reasons, I must recommend rejection of the manuscript at this time.

Author Response

Dear Reviewer, with respect to your Comments and Suggestions for Authors:

We appreciate your feedback about our work

We also appreciate your suggestions to improve it.

All tables have the general structure depicted in scheme 1 (BTs)

All the typing mistakes were corrected

The numbering was corrected

Reviewer 2 Report

Comments and Suggestions for Authors

Overall, a potentially useful review. However, there are several grammatical and punctuation mistakes scattered throughout the text. These make the reading very uneasy. Major revisions are necessary due to the quality of English, but the review has value as a reference for researched interested in the synthesis of BT derivatives.

Some limited examples of ENG mistakes are:

e.g. "bibiographyc" in abstract and line 72

line 39 "...however some of them has been found..."

line 90: "...Compound 1g were the..."

line 105: "...proposed as powered candidates..."

line 109 and 351, 469 "temperatura"

line 236: "...co-friendly fast and, efficient..." (fast, and efficient = comma before and)

line 265: "...avoided in agree with the green chemistry..."

line 396: "...Under alkaline conditions, the probe shown cyan blue fluorescence..."

line 433: "...compound 57e had a promised value..."

line 518: "...mice treated with a doce..."

line 738: "literatura"

etc.

Is a "Results" section appropriate for a review? 

I suggest deleting sentences of the type "molecules were characterized by NMR" etc. Authors should be able to make a critical screening and decision to include studies they deem correct.

Scheme 5: I'd suggest adding in the scheme caption the reaction conditions A, B, C and D rather than describing them in the text.

Definition of abbreviations or acronyms are missing. What is e.g. SPF? of UVA-PF? (line 93). Other examples are found later in the text.

Line 300: there is an hyperlink associated with "ethyl"

Comments on the Quality of English Language

Mid-level. As mentioned in the comments, a careful and significant editing of the manuscript is necessary.

Author Response

Dear reviewer

We Thanks a lot your opinions and comments about our article review. On this base, we reestructured the article and refined the english language as well as the discusión.

We hope that the corrections meet the required expectations

Reviewer 3 Report

Comments and Suggestions for Authors

This review by Cruz et al. describes an extensive study on the synthesis of the aromatic bicyclic ring system benzothiazole (BT), which is a type of structure widely present in interesting natural products with important biological activities and is used in medicinal chemistry. 

The authors described an extense variety of synthetic processes for the preparation of several compounds, including BT skeletons. In addition, the authors have provided interesting information about the biological activity detected in this type of compounds.  

This review constitutes a good attempt to compile relevant information on bioactive benzothiazole’s type compounds with potential therapeutic activity against several diseases, such as cancer or Alzheimer. The authors provide the scientific community with a good compilation of data that can be used in the future by other authors to develop new compounds with related structures and improved activities. 

However, this review has significant conceptual and editorial flaws that require improvements. Some issues, such as writing errors in the names of compounds or diseases, the use of capital letter for the name of chemical compounds (this is not necessary) and the existence of multiple errors in the English writing (even in one of the e-mail addresses of the corresponding author), make reading this review difficult. In addition, the number of compounds included in the Schemes is excessive. In a revision article, it is enough to include one or two examples of the synthesized compounds or a general structure, and not all the described products in the corresponding article. For example, Scheme 56 is completely incomprehensible. Several mistakes have been detected in the Schemes and Figures. On the other hand, in some parts of the manuscript, some words with links associated to any IA have been found. Other problems are the heading of the different sections in the manuscript, without the corresponding section number. Finally, information required in the template of IJMS have not been provided or removed, such as “Author Contributions”, “Funding” or “Conflicts of Interest”. 

For these reasons, I consider that this article is not yet ready for publication, and more extensive revision and improvements are necessary. Therefore, I recommend that this article be rejected at its current state and that the authors resubmit it after making significant refinements.

Comments on the Quality of English Language

I have detected several mistakes. They could be only typing errors, but an extensive revision is required. 

Author Response

(The authors gave the same response as above.)

Round 2

Reviewer 1 Report

Comments and Suggestions for Authors

Dear Authors,

Thank you for submitting your revised manuscript. I appreciate your efforts in addressing the previous feedback. I have carefully reviewed the updated version, and I acknowledge the improvements made, particularly regarding the abstract and certain aspects of data presentation.

I am pleased to note that the Abstract has been substantially improved and now provides a more comprehensive overview of the review's content. The conversion of many schemes into tables is also a positive change in terms of organization and clarity of data presentation. However, my primary concerns, previously outlined as "Major Concerns" regarding the predominantly descriptive approach and the lack of in-depth critical analysis, remain largely unaddressed. While the manuscript is a valuable compilation of recent literature on 2-substituted benzothiazoles, it still presents synthetic methods sequentially without sufficient comparative discussion or deeper critical analysis.

For instance, the review would be significantly strengthened by:

Meaningful Comparison of Methods: A more explicit and detailed comparison of the relative efficiencies, substrate scopes, inherent limitations, environmental impact ("greenness"), and potential cost-effectiveness of the various methods, catalysts, and conditions discussed. Simply stating the advantages of a given method, as reported in the original source, does not constitute a comprehensive critical analysis across multiple studies.

Deeper Rationalization of Trends: The discussion of substituent effects on reaction outcomes or biological activity still appears to primarily summarize findings from individual primary sources, rather than synthesizing broader trends or providing deeper rationalization across multiple studies. This limits the critical guidance offered to the reader beyond the compilation itself.

As currently presented, the descriptive style makes it challenging for the reader to effectively discern the specific advantages of certain methods or to select appropriate strategies for particular synthetic targets or substrate types. To further enhance the manuscript's scientific impact, I strongly recommend that you dedicate specific sections or paragraphs to an in-depth critical analysis. This would involve actively comparing and contrasting the reported methods, highlighting their strengths and weaknesses in a more analytical rather than purely descriptive manner.

Thank you once again for your revisions. I look forward to reviewing a version that incorporates a more robust critical analysis.

Author Response

Dear Reviewer, with respect to your Comments and Suggestions for Authors:

We appreciate your feedback about our work

We also appreciate your suggestions to improve it.

To further enhance the manuscript's scientific impact:

We addressed a more descriptive approach and more deeper critical analysis

We placed the yields in the tables

In the text we discuss some effects of substituent groups on benzothiazoles, with respect to yielings and biological activity.

We included in the conclutions section a comparative discussion about the differente synthesis methods, including a table of new catalysts used recently

Reviewer 2 Report

Comments and Suggestions for Authors

An overall improved manuscript. As previously suggested, in my opinion, the review can be considered for publication, after text editing. I do appreciate the difficulty of writing in ENG when this is not the mother toungue, but some editing is still needed (staring from the abstract).

(There are online services and free tools that could help the authors.)

Comments on the Quality of English Language

Some editorial work is still needed as the paper reads awkward in some instances. 

Author Response

Dear Reviewer, with respect to your Comments and Suggestions for Authors:

We appreciate your feedback about our work

We also appreciate your suggestions to improve it.

We made an effort to edit the text regarding the English language, we hope it is more readable.

Reviewer 3 Report

Comments and Suggestions for Authors

This review by Cruz et al. describes an extensive study on the synthesis of the aromatic bicyclic ring system benzothiazole (BT), which is a type of structure widely present in interesting natural products with important biological activities and is used in medicinal chemistry. 

The authors described an extensive variety of synthetic processes for the preparation of several compounds, including BT skeletons. In addition, the authors have provided interesting information about the biological activity detected in this type of compounds.

As I indicated in my previous report, I think that this review represents a good attempt to provide interesting information on bioactive benzothiazole’s type compounds with potential therapeutic activity against several diseases, such as cancer and Alzheimer’s disease. The authors provide the scientific community with a good compilation of data that can be used in the future by other authors to develop new compounds with related structures and improved activities. 

However, although the authors have included some required improvements in the manuscript, several problems have been identified. 

  1. Several typing mistakes are still present in the manuscript. For example, in Figure 2, the name of compound GW610,NSC721648 is incorrect, and the subscripts have been omitted. On page 2, line 66, there is no space between “agents” and “[51-58…”. On page 3, line 75, the reference is located after the point, not before. These are only a few examples of typing mistakes that can be found in all of the manuscript.
  2. On page 3, lines 84-85, the authors introduce compounds 1 and 2. However, on page 4, line 94, compounds 21, 22, and 23 are mentioned. What about the rest of compounds from 3 to 20? 
  3. On page 4, a really strange Table 1 is shown. I do not understand this table. The structures of the indicated compounds are unknown. If the authors wish to depict a collection of similar compounds, it would be necessary to indicate a general structure. 
  4. The same problem appears in the rest of the tables depicted by the authors. For example, in Table 2… who is compound 24? Or 25 in table 3? In my opinion, the authors want to include several compounds, but this is not necessarily. In a revision article, it is enough to include one or two examples of the synthesized compounds or a general structure.
  5. Later, on page 5, the authors mentioned compound 29 (also unknown), and suddenly, compounds 210, 211, and 212 appeared. What’s about the rest? A very important problem in the numeration of compounds mentioned in the article is present in the manuscript. 

  For these reasons, my first recommendation is also valid for this improvement attempt. I consider the information provided in this manuscript interesting and valid for publication in IJMS, but the current state of the manuscript requires much more improvement. I recommend that the manuscript be accepted only after an extensive MAJOR REVISION.

Author Response

(The authors gave the same response as above.)

Round 3

Reviewer 1 Report

Comments and Suggestions for Authors

Dear Authors,

Thank you for submitting the revised version of your manuscript. I've carefully reviewed the updated submission and can confirm that you've diligently addressed the concerns raised in my previous feedback.

Specifically, I appreciate the significant efforts made to enhance the critical analysis and comparative discussion of the synthetic methods. The inclusion of more in-depth insights into the relative efficiencies, substituent effects, and advantages/disadvantages of various approaches, as well as the expanded discussion in the conclusions, substantially improves the scientific value and impact of the review. The improvements in the abstract are also noted and contribute to a more comprehensive summary of the work.

These revisions have effectively transformed the manuscript from a predominantly descriptive compilation into a more analytical and insightful review, aligning it with the expected standards for publication in the International Journal of Molecular Sciences.

Therefore, I'm happy to recommend the acceptance of your manuscript in its current form.

Reviewer 3 Report

Comments and Suggestions for Authors

The authors have included the improvements and corrections requested in my previous reviews.
I recommend accepting the article in its present form.